# PromptGNN-sim: Deep Fusion and Alignment of GNN and LLMs for Text-Attributed Graph Learning

## Abstract

Text-Attributed Graphs (TAGs), which integrate rich textual semantics with graph structural information, play a critical role in graph learning tasks. However, current fusion approaches suffer from a fundamental limitation: they treat textual and structural modalities as separate inputs in a shallow, unidirectional pipeline. This one-way information flow prevents a deep, interactive exchange between modalities, leading to suboptimal performance, particularly in challenging scenarios with sparse connectivity and when generalizing across different graphs. To overcome these limitations, we introduce PromptGNN-sim, a novel bi-directional structure-semantic fusion framework that enables deep, symbiotic collaboration between GNNs and LLMs. At its core, PromptGNN-sim leverages a Graph Attention Network (GAT) to perform semantically-aware neighborhood selection, combining structural attention with textual similarity. This GNN-derived structural context is then used to dynamically generate rich, structure-aware prompts for an LLM, which explicitly include the target node's textual summary, label categories, and representative keywords from semantically similar neighbors. Unlike traditional methods, our framework incorporates bi-directional cross-modal contrastive learning and cross-attention mechanisms during training to jointly optimize both GNN and LLM components for enhanced performance and robustness. We conduct comprehensive experiments on six public datasets, including Cora, Pubmed, and WikiCS, evaluating both task performance and robustness under cross-task transfer, cross-dataset generalisation, and sparse perturbations. Results show that PromptGNN-sim significantly outperforms classical GNNs, LLMs, and recent state-of-the-art GNN–LLM fusion methods in terms of accuracy, generalisation, and robustness. This work not only introduces an effective framework for deep GNN–LLM collaboration but also lays a solid foundation for future research on truly interactive multi-modal graph learning.

## 1 Introduction

Text-Attributed Graphs (TAGs) have become a key paradigm for modeling complex relational data enriched with rich textual semantics. In TAGs, each node is associated with unstructured text describing its attributes, while edges capture diverse, often heterogeneous relationships representing various interactions or dependencies. Such graphs are common in domains like academic citation networks Giles et al. (1998); Yan et al. (2023), where publications are connected by citations and described by abstracts; social media platforms Zhou et al. (2020); Hu et al. (2020), reflecting user interactions alongside textual content such as posts and comments; and product knowledge bases Jin et al. (2024), which integrate product relations with descriptive reviews. By jointly leveraging graph topology and textual information, TAGs enable effective representation learning for downstream tasks including node classification, link prediction, and recommendation Hamilton et al. (2017); Veličković et al. (2017); Zhang et al. (2025). This combined modeling exploits the complementary strengths of graph structure—providing relational inductive biases Kipf (2016); Xu et al. (2018)—and textual semantics, which offer dense, high-dimensional contextual cues.

However, effectively integrating structural and textual modalities in TAGs remains challenging due to their intrinsic differences and the complexities of real-world data Yan et al. (2023). Graph con-

nectivity encodes discrete relational patterns that are often sparse, incomplete, or noisy—such as missing citations in academic networks or spurious links on social platforms—reducing the reliability of purely structural signals Wu et al. (2020). Textual attributes also vary substantially in quality, style, and length across nodes and domains, complicating semantic understanding and transfer Li et al. (2024b); Wang et al. (2025a). Existing approaches predominantly employ shallow fusion techniques, like embedding concatenation or late fusion, which treat modalities independently and limit the learning of deep interactive representations Zhu et al. (2024); Jin et al. (2024). Such simplistic strategies often underperform in realistic settings characterized by noisy inputs, sparse connectivity, and domain shifts Zhang et al. (2025); Li et al. (2024b), resulting in degraded robustness, generalization, and adaptability.

In addition, neighborhood aggregation typically depends solely on structural adjacency, neglecting semantic relevance Li et al. (2024a), which further weakens model resilience in dynamic or corrupted graphs Wu et al. (2020); Dai et al. (2018). The challenges are compounded by the dynamic nature of many TAGs, whose structure and textual content evolve over time Rossi et al. (2020); Zheng et al. (2025). This evolution demands fusion frameworks that are adaptive and robust to temporal changes Pareja et al. (2020); Roy et al. (2025). Critically, systematic investigations into how fusion strategies impact model robustness, generalization, and resistance to perturbations such as data sparsity, adversarial noise, and domain shifts remain limited Dai et al. (2018); Wang et al. (2025a); Li et al. (2024b). Addressing these gaps requires unified frameworks that enable deep, bidirectional fusion of graph structural and textual information, fully leveraging their complementary strengths Zhang et al. (2025).

To address these challenges, we propose **PromptGNN-SIM**, a framework enabling deep, bidirectional fusion of graph structure and textual semantics via collaboration between GNNs and LLMs. At its core, a Graph Attention Network (GAT) integrates structural attention with textual similarity for semantically-aware neighbor selection. This structural context dynamically guides prompt generation for the LLM, incorporating node summaries, label categories, and keywords from relevant neighbors. Unlike prior methods, we jointly optimize GNN and LLM components using contrastive learning and cross-attention to achieve robust, interactive alignment. We focus on three research questions(**RQs**):(1)How can graph structure and semantic similarity be effectively incorporated into dynamic prompt generation for early-stage modality fusion?(2) How can bi-directional cross-modal attention be designed to facilitate rich interactions between textual semantics and graph structure for improved node representations?(3)How does the proposed framework enhance robustness and generalization across different real-world scenarios?Based on these questions, we highlight our main contributions:

- **Dynamic prompting mechanism.**We propose a structure-aware dynamic prompting framework that adaptively integrates node texts with selectively filtered neighborhood information based on semantic similarity and structural attention, enabling effective early fusion for node classification and link prediction on text-attributed graphs.

- **Bi-directional cross-modal attention.** We design a dual attention module that facilitates mutual interaction between textual and structural modalities, effectively capturing complementary signals and enhancing node representations.

- **Contrastive learning for prompt-text alignment.** We introduce a multi-view contrastive objective to align raw node texts with dynamically generated prompts, encouraging robust and view-invariant semantic representations.

- **Extensive evaluation on robustness and transferability.** Experiments on six real-world datasets demonstrate the framework's strong generalization across domains and robustness to structural noise and input perturbations, validating its effectiveness under diverse and challenging conditions.

## 2 FORMALIZATION

In this section, we provide formal definitions of TAGs, describe two key tasks—node classification and link prediction—and summarize core modeling principles based on large language models and graph neural networks on TAGs.

**Text-Attributed Graphs.** We study Text-Attributed Graphs (TAGs), defined as $\mathcal{G} = (\mathcal{V}, \mathcal{E}, \mathcal{X})$, where $\mathcal{V}$ is the node set, $\mathcal{E}$ the edge set, and $\mathcal{X}$ the associated node texts. Each node $v_i$ has a token sequence $\mathbf{X}_i$. While $\mathcal{E}$ captures graph structure, $\mathcal{X}$ encodes semantic content. We address node classification, learning $f : \mathcal{V} \to \mathcal{Y}$ from partial labels, and link prediction, estimating $s : \mathcal{V} \times \mathcal{V} \to \mathbb{R}$ to infer edges. Joint modelling of structure and text remains challenging.

**Graph Neural Networks.** Graph Neural Networks (GNNs) follow a message-passing framework where node representations $\mathbf{h}_i^{(k)}$ at layer $k$ are updated by aggregating neighbor information and applying a nonlinear transformation $\sigma(\cdot)$ Kipf (2016); Wu et al. (2020).

$$\mathbf{h}_i^{(k)} = \sigma \left( \text{AGGREGATE} \left( \left\{ \mathbf{h}_j^{(k-1)} : j \in \mathcal{N}(i) \right\} ; \mathbf{W}^{(k)} \right) \right), \tag{1}$$

Here, $\mathbf{h}_j^{(k-1)}$ is the neighbor $j$'s feature from the previous layer, $\mathbf{W}^{(k)}$ a learnable weight matrix, and $\mathcal{N}(i)$ the neighbors of node $i$. The aggregation function varies across GNNs, including mean, sum, max pooling,etc.Hamilton et al. (2017); Xu et al. (2018); Veličković et al. (2017).

**Large Language Models (LLMs) and Language Models (LMs).** For each node $i$ with textual attribute $x_i$, we first generate a textual description $t_i = \text{LLM}(x_i)$ using a large language model (LLM) via prompting. This description is then encoded into a dense semantic vector $z_i = \text{LM}(t_i)$ by a language model (LM), producing embeddings suitable for downstream graph learning.

**Fusion with GNNs.** GNNs aggregate information from neighboring nodes to capture structural context. To leverage both the structural information from GNNs and the semantic embeddings from the LM, we introduce a learnable fusion operator:

$$h_i^{\text{fused}} = \text{Fusion}(h_i, z_i), \tag{2}$$

where $h_i$ is the node representation from GNNs. The fusion can be implemented via concatenation, gating, or attention-based mechanisms, allowing the model to capture complementary structural and semantic information.

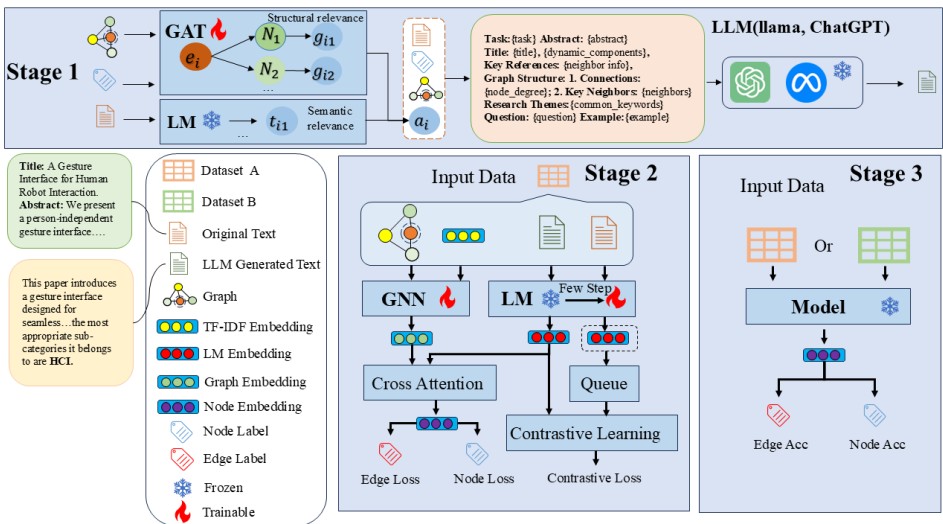

Figure 1: The architecture of PromptGNN-sim. The framework first generates textual descriptions for nodes using LLMs, converts them into semantic embeddings via LMs, and fuses them with GNN-based structural embeddings through a learnable fusion module.

## 3 METHODOLOGY

In this section, we present **PromptGNN-sim**, a unified framework for node classification and link prediction on text-attributed graphs (TAGs). As illustrated in Figure 1, our framework consists of

three main components: 1) a dynamic prompting mechanism that synergistically synthesizes information from node texts, graph topology, 2) a cross-attention fusion module designed to seamlessly integrate rich textual semantics with graph structural representations, and 3) a contrastive learning objective to enforce semantic alignment between the original and predicted texts.

## 3.1 DYNAMIC PROMPT CONSTRUCTION

To provide LLMs with comprehensive, context-aware inputs for node classification and link prediction, we propose a dynamic prompt framework. This component addresses **RQ1** by effectively incorporating graph structure and semantic similarity into prompt generation for early-stage fusion. Our approach adaptively integrates intrinsic node texts with relational context, balancing semantic and structural information to enhance representation learning.

**Feature Representation.** We first obtain two distinct feature representations to capture the node's semantic and structural properties. For a text node $T_i$ with $n_i$ tokens, we leverage a pre-trained language model (e.g., BERT) to acquire contextual token embeddings $h_j^{(i)} \in \mathbb{R}^d$. The overall semantic representation of the node, $v_i$, is then derived by mean pooling over these embeddings, which captures rich contextual semantics beyond static word representations:

$$v_i = \frac{1}{n_i} \sum_{j=1}^{n_i} h_j^{(i)}. \tag{3}$$

Simultaneously, we employ a Graph Attention Network (GAT) to encode structural information. The GAT computes a learnable attention weight $\alpha_{v,u}$ for each neighbor $u$ of node $v$, reflecting its structural importance within the graph:

$$\alpha_{v,u} = \frac{\exp\left(\text{LeakyReLU}\left(\mathbf{a}^\top[\mathbf{W}\mathbf{x}_v \parallel \mathbf{W}\mathbf{x}_u]\right)\right)}{\sum_{k \in \mathcal{N}(v)} \exp\left(\text{LeakyReLU}\left(\mathbf{a}^\top[\mathbf{W}\mathbf{x}_v \parallel \mathbf{W}\mathbf{x}_k]\right)\right)}. \tag{4}$$

**Adaptive Fusion for Neighbor Selection.** We introduce an adaptive fusion mechanism to balance the influence of structural and semantic cues based on the node's textual content. For a node $v$ and its neighbor $u$, the fused weight $w_{v,u}$ is a linear combination of the structural attention weight $\alpha_{v,u}$ and the semantic cosine similarity $s_{v,u}$ ($s_{v,u} = \text{sim}(v_v, v_u)$). The fusion coefficient $\lambda_v$ is dynamically determined by the node's text length $n_v$ relative to a threshold $L$:

$$w_{v,u} = \lambda_v \alpha_{v,u} + (1 - \lambda_v)s_{v,u}, \quad \text{where} \quad \lambda_v = \begin{cases} \lambda, & n_v < L, \\ 1 - \lambda, & n_v \geq L, \end{cases} \tag{5}$$

where $\lambda \in (0, 1)$ is a hyperparameter. This approach ensures that shorter texts rely more on structural cues, while longer texts leverage their rich semantic content. We then select the top-$k$ neighbors with the highest fused weights to form a filtered and contextually relevant neighborhood.

**Dynamic Prompt Engineering.** The final step is to translate these insights into a structured, natural language prompt for the LLM. We design two types of prompts based on the node's characteristics, which are then concatenated. The structural prompt $P_{\text{struct}}(v)$ guides the LLM on how to utilize the neighborhood context based on the node's degree $d_v$:

$$P_{\text{struct}}(v) = \begin{cases} \text{"This paper is not highly cited. Prioritize its intrinsic content."} & d_v < d_{\text{th}}, \\ \text{"This paper is highly cited. Synthesize information from its influential neighbors."} & d_v \geq d_{\text{th}}, \end{cases} \tag{6}$$

The semantic prompt $P_{\text{sem}}(v)$ instructs the LLM on how to interpret the provided textual data, based on its length $n_v$:

$$P_{\text{sem}}(v) = \begin{cases} \text{"The full text is provided. Focus on methodological details and experimental results."} & n_v \geq l, \\ \text{"The abstract is provided. Focus on the core contributions and research problem."} & n_v < l, \end{cases} \tag{7}$$

Here, the threshold $l$ in the semantic prompt differs from the text-length threshold $L$ used in Eq. (5), as they control different mechanisms. The final prompt $P(v) = P_{\text{struct}}(v) \oplus P_{\text{sem}}(v)$ is then combined with the filtered neighborhood information, providing a dynamic and powerful input for the LLM to perform accurate node classification.

## 3.2 Cross-Modal Attention Fusion

To effectively capture the deep semantic associations between textual information and graph structures, we propose a **bi-directional cross-modal attention mechanism**. This mechanism facilitates information exchange between text and graph representations through two independent attention modules, addressing **RQ2** by enabling rich interactions for improved node representations.

Given the textual embeddings $\mathbf{T} \in \mathbb{R}^{B \times L \times d_t}$ and graph-based neighbor representations $\mathbf{G} \in \mathbb{R}^{B \times K \times d_g}$, we first project them into a shared latent space of dimension $d_h$:

$$\mathbf{T}' = \mathbf{T}W_t, \quad \mathbf{G}' = \mathbf{G}W_g, \tag{8}$$

where $W_t \in \mathbb{R}^{d_t \times d_h}$ and $W_g \in \mathbb{R}^{d_g \times d_h}$ are learnable linear projection matrices.

The first module, **Text-guided Graph Attention**, uses the projected text embeddings $\mathbf{T}'$ as the query to aggregate context from the graph representations $\mathbf{G}'$ which serve as the key and value. The resulting attended features $\mathbf{A}_{tg}$ are then mean-pooled to form a single representative vector $\mathbf{z}_g$:

$$\mathbf{A}_{tg} = \text{MultiHeadAttention}(\mathbf{Q} = \mathbf{T}', \mathbf{K} = \mathbf{G}', \mathbf{V} = \mathbf{G}') \tag{9}$$

$$\mathbf{z}_g = \text{MeanPool}(\mathbf{A}_{tg}) \tag{10}$$

Conversely, the second module, **Graph-guided Text Attention**, aggregates information from the text. We first obtain a comprehensive graph query vector $\mathbf{q}_g$ by applying mean pooling to the graph representations $\mathbf{G}'$. This vector then serves as the query to guide attention over the text features $\mathbf{T}'$, which act as the key and value:

$$\mathbf{q}_g = \text{MeanPool}(\mathbf{G}') \tag{11}$$

$$\mathbf{z}_t = \text{MultiHeadAttention}(\mathbf{Q} = \mathbf{q}_g, \mathbf{K} = \mathbf{T}', \mathbf{V} = \mathbf{T}') \tag{12}$$

The mechanism produces two distinct, enhanced representations, $\mathbf{z}_t$ and $\mathbf{z}_g$, which can be used for downstream tasks. Dropout is applied to each vector independently to improve the model's generalization.

## 3.3 Contrastive Learning

To enhance the semantic richness and robustness of our node representations, we introduce a contrastive learning objective. This objective operates by aligning two distinct textual "views" of each node: the original, unprocessed text and a structured summary generated by a Large Language Model (LLM). This process encourages the model to learn view-invariant features, focusing on the core semantic content of the node.

For each node, we generate two distinct textual embeddings. The first, $\mathbf{t}_{\text{raw}}$, is derived from the node's original raw text description. The second, $\mathbf{t}_{\text{sum}}$, is derived from a structured analysis generated by an LLM. This analysis is comprehensive, containing not only the LLM's classification prediction for the node but also a detailed rationale explanation for this prediction.

Both views are first encoded using a shared text encoder and then projected into a common latent space using separate linear projection heads, a crucial component popularized by SimCLR Chen et al. (2020). This yields the final representations $\mathbf{z}_{\text{raw}}$ and $\mathbf{z}_{\text{sum}}$, respectively.

We employ a symmetric InfoNCE loss function Oord et al. (2018) using memory queues He et al. (2020) to provide a large set of negative samples. The final contrastive learning objective is defined as:

$$\mathcal{L}_{\text{contrast}} = -\frac{1}{2} \left( \log \frac{\exp(\text{sum}(\mathbf{z}_{\text{raw}}, \mathbf{z}_{\text{sum}})/\tau)}{\exp(\text{sum}(\mathbf{z}_{\text{raw}}, \mathbf{z}_{\text{sum}})/\tau) + \sum_{k=1}^{K} \exp(\text{sum}(\mathbf{z}_{\text{raw}}, \mathbf{q}_k)/\tau)} \right.$$
$$\left. + \log \frac{\exp(\text{sum}(\mathbf{z}_{\text{sum}}, \mathbf{z}_{\text{raw}})/\tau)}{\exp(\text{sum}(\mathbf{z}_{\text{sum}}, \mathbf{z}_{\text{raw}})/\tau) + \sum_{k=1}^{K} \exp(\text{sum}(\mathbf{z}_{\text{sum}}, \mathbf{p}_k)/\tau)} \right) \tag{13}$$

where $\text{sim}(\cdot, \cdot)$ is the cosine similarity, and $\tau$ is a temperature parameter Chen et al. (2020). The terms $\mathbf{q}_k$ and $\mathbf{p}_k$ represent negative embeddings for the analysis and raw text views, respectively, which are sampled from two corresponding memory queues, $\mathcal{Q}_{\text{analysis}}$ and $\mathcal{Q}_{\text{raw}}$.

This loss is scaled by a regularization hyperparameter $reg$ and added to the task loss.

## 4 EXPERIMENTS

We conduct extensive experiments on multiple real-world text-attributed graph datasets to validate the effectiveness and generalisation ability of the proposed method, PromptGNN-sim.

### 4.1 COMPARATIVE RESULTS

This subsection compares our method with various baselines on node classification and link prediction benchmarks. Results demonstrate that integrating GNNs with LLMs consistently improves performance across datasets and tasks. Quantitative analysis confirms our approach's superiority in capturing complex structural and semantic information for enhanced predictions.

Table 1: Comparison of Node Classification Accuracy (%)

| Category | Models | CORA | PUBMED | CITESEER | WIKICS | HISTORY | PHOTO |
|---|---|---|---|---|---|---|---|
| GNNs | GCN | 85.23 ± 0.32 | 82.98 ± 0.08 | 73.51 ± 0.25 | 80.52 ± 0.19 | 77.70 ± 0.41 | 77.39 ± 0.22 |
| | MLP | 74.16 ± 0.12 | 88.08 ± 0.07 | 79.15 ± 0.40 | 79.88 ± 0.48 | 82.55 ± 0.14 | 73.98 ± 0.34 |
| | GraphSAGE | 85.42 ± 0.46 | 82.78 ± 0.06 | 74.13 ± 0.33 | 79.96 ± 0.37 | 80.11 ± 0.17 | 81.67 ± 0.28 |
| | GAT | 85.79 ± 0.28 | 80.90 ± 0.05 | 74.45 ± 0.41 | 81.24 ± 0.35 | 81.35 ± 0.46 | 83.73 ± 0.29 |
| | NodeFormer | 77.85 ± 0.13 | 79.91 ± 0.06 | 73.04 ± 0.21 | 80.64 ± 0.38 | 77.36 ± 0.37 | 75.26 ± 0.27 |
| | GLNN | 86.16 ± 0.15 | 81.97 ± 0.10 | 72.72 ± 0.40 | 80.73 ± 0.42 | 78.77 ± 0.26 | 84.54 ± 0.44 |
| | GraphCL | 86.53 ± 0.33 | 82.98 ± 0.06 | 73.66 ± 0.39 | 79.19 ± 0.27 | 77.15 ± 0.25 | 77.57 ± 0.32 |
| | Graphormer | 83.39 ± 0.39 | 77.89 ± 0.10 | 71.94 ± 0.11 | 76.84 ± 0.44 | 73.32 ± 0.18 | 80.35 ± 0.30 |
| BERT | BERT | 72.27 ± 0.23 | 90.25 ± 0.08 | 75.65 ± 0.43 | 80.35 ± 0.31 | 83.45 ± 0.36 | 73.93 ± 0.28 |
| SOTA | ENGINE | 88.34 ± 0.40 | 92.18 ± 0.06 | 78.24 ± 0.13 | 81.04 ± 0.33 | 84.52 ± 0.25 | 83.12 ± 0.41 |
| | ULTRATAG-S | 90.96 ± 0.45 | 92.41 ± 0.30 | 78.68 ± 0.21 | 83.05 ± 0.16 | – | 84.70 ± 0.03 |
| | OFA | 74.76 ± 1.22 | 78.25 ± 0.17 | – | 77.65 ± 0.22 | – | – |
| | GraphPrompter | 80.26 | 94.80 | 73.61 | 80.98 | 79.42 | 80.04 |
| | PromptGFM (Flan-T5) | 91.72 | 92.83 | 84.49 | 81.49 | 82.33 | 85.41 |
| | PromptGFM (Llama3) | 92.42 | 94.65 | 85.32 | 84.66 | 86.72 | 86.61 |
| Ours | PromptGNN-sim (Llama3B) | 90.59 ± 0.33 | 94.12 ± 0.09 | 82.76 ± 0.22 | 85.82 ± 0.44 | 84.97 ± 0.26 | 87.20 ± 0.18 |

Table 2: Comparison of Link Prediction Accuracy (%)

| Category | MODELS | CORA | PUBMED | CITESEER | WIKICS | HISTORY | PHOTO |
|---|---|---|---|---|---|---|---|
| GNNs | GCN | 77.65 ± 0.31 | 76.08 ± 0.25 | 77.87 ± 0.42 | 78.40 ± 0.19 | 80.54 ± 0.28 | 79.13 ± 0.35 |
| | MLP | 75.99 ± 0.45 | 74.83 ± 0.33 | 77.90 ± 0.29 | 73.24 ± 0.51 | 77.27 ± 0.37 | 70.14 ± 0.48 |
| | GraphSAGE | 79.99 ± 0.22 | 75.26 ± 0.41 | 81.12 ± 0.18 | 76.43 ± 0.36 | 78.63 ± 0.33 | 78.33 ± 0.29 |
| | GAT | 71.33 ± 0.53 | 70.15 ± 0.48 | 72.06 ± 0.50 | 74.12 ± 0.41 | 76.59 ± 0.38 | 77.98 ± 0.40 |
| | NodeFormer | 61.48 ± 0.61 | 58.08 ± 0.55 | 63.06 ± 0.49 | 62.04 ± 0.58 | 63.20 ± 0.51 | 60.54 ± 0.63 |
| | GLNN | 69.91 ± 0.44 | 72.59 ± 0.39 | 71.88 ± 0.47 | 70.41 ± 0.42 | 72.12 ± 0.35 | 74.65 ± 0.31 |
| | GraphCL | 77.01 ± 0.29 | 76.62 ± 0.31 | 75.51 ± 0.35 | 77.44 ± 0.27 | 80.20 ± 0.24 | 80.50 ± 0.22 |
| | Graphormer | 68.41 ± 0.50 | 63.68 ± 0.62 | 62.24 ± 0.54 | 68.40 ± 0.48 | 69.83 ± 0.45 | 69.15 ± 0.51 |
| BERT | BERT | 60.53 ± 0.58 | 89.54 ± 0.11 | 93.19 ± 0.08 | 89.32 ± 0.15 | 92.98 ± 0.09 | 89.37 ± 0.13 |
| | Sentence-BERT | 78.97 ± 0.21 | 90.33 ± 0.10 | 85.73 ± 0.18 | 91.75 ± 0.09 | 86.76 ± 0.16 | 79.99 ± 0.25 |
| Ours | PromptGNN-sim (Llama3B) | 87.54 ± 0.15 | 88.30 ± 0.13 | 86.27 ± 0.19 | 89.93 ± 0.12 | 89.57 ± 0.14 | 89.55 ± 0.11 |

Table 1 presents node classification results on six benchmark datasets. Traditional GNNs (e.g., GCN, GraphSAGE, GAT) deliver stable performance across most datasets, with GAT and GraphCL being competitive in citation networks, while MLP achieves strong results on PubMed due to high feature separability. Pre-trained language models such as BERT and RoBERTa excel on semantically rich datasets like PubMed and History, highlighting the benefit of contextualized embeddings. Recent SOTA methods (ENGINE, ULTRATAG-S) further improve accuracy on several datasets. In addition, we incorporate three strong LLM-based SOTA GNN baselines—GraphPrompter, PromptGFM

Table 3: Zero-shot Transfer Learning from Node Classification to Link Prediction

| DATASETS | TASK | AUC↑ | AP↑ | F1↑ | ACC↑ |
|---|---|---|---|---|---|
| Cora | NC → LP | 90.70 ± 0.31 | 91.43 ± 0.30 | 82.89 ± 0.28 | 82.89 ± 0.25 |
| | NC → NC | 95.10 ± 0.21 | 95.08 ± 0.18 | 87.51 ± 0.22 | 87.54 ± 0.20 |
| Citeseer | NC → LP | 94.99 ± 0.25 | 95.15 ± 0.21 | 88.34 ± 0.23 | 88.34 ± 0.24 |
| | NC → NC | 95.11 ± 0.19 | 94.91 ± 0.17 | 86.16 ± 0.20 | 86.27 ± 0.18 |
| PubMed | NC → LP | 89.93 ± 0.33 | 90.87 ± 0.27 | 81.69 ± 0.31 | 81.69 ± 0.28 |
| | NC → NC | 95.21 ± 0.24 | 95.25 ± 0.22 | 87.90 ± 0.26 | 88.30 ± 0.23 |
| Wikics | NC → LP | 96.32 ± 0.29 | 96.52 ± 0.24 | 88.98 ± 0.27 | 89.31 ± 0.25 |
| | NC → NC | 98.48 ± 0.19 | 98.39 ± 0.17 | 90.94 ± 0.21 | 89.94 ± 0.19 |

(Flan-T5), and PromptGFM (Llama3)—into Table 1, which are shown in blue to indicate newly added baselines, and our PromptGNN-sim achieves competitive performance overall, outperforming these baselines on WikiCS and Photo while remaining comparable on the remaining datasets. Notably, a subsequent run achieves 91.14% accuracy (91.04% macro-F1) on Cora, further widening the margin. For consistency with the large number of previously completed ablation studies, we report the earlier result in Table 1, but note that the latest run demonstrates even stronger performance on cora, underscoring the robustness and generalization capability of our approach across diverse graph domains. Moreover, the strong performance on the medium-scale and information-rich Photo dataset (48k nodes, 500k edges) further demonstrates that our architecture maintains both efficiency and effectiveness as graph size increases, providing empirical support for its scalability toward even larger OGBN-scale graphs.

Table 2 presents the experimental results across six benchmark datasets.Traditional graph neural networks (GNNs) demonstrate stable and consistent performance, whereas pre-trained language models exhibit superior results on datasets characterized by rich semantic content, such as PubMed and CiteSeer. Notably, our proposed hybrid model, combining GNN with a large language model (Llama3B), attains the highest accuracy on all evaluated datasets. It achieves substantial improvements, with gains reaching up to 9.89% over the strongest GNN baseline on the Cora dataset. These findings underscore the efficacy of integrating structural graph information with semantic knowledge encapsulated by large language models. Furthermore, the consistent performance enhancements observed across heterogeneous datasets underscore the robust generalization capability of our approach in link prediction tasks.

### 4.2 Transferability Across Tasks and Domains

This subsection investigates the transferability of our proposed model across different tasks, domains, and graph datasets. In line with **RQ3**, we evaluate how the framework enhances robustness and generalization beyond the original training conditions. Experimental results demonstrate that integrating graph neural networks with large language models effectively captures transferable representations, enabling consistent performance gains across heterogeneous graphs and diverse scenarios.

Table 3 presents the model's performance under a cross-task setting, where node classification (NC) serves as the training task and link prediction (LP) as the test task. The results indicate that representations learned from node classification effectively transfer to link prediction, achieving robust performance across all datasets. Although cross-task performance is slightly lower than the within-task (NC→NC) results, the marginal degradation demonstrates strong generalization of the learned features. Consistently high AUC and AP scores across diverse datasets further validate the method's capability to capture both structural and semantic information, highlighting the potential of multi-task and transfer learning frameworks in graph representation learning.

Table 4 evaluates the cross-domain generalization of models trained on the CORA dataset for node classification across Citeseer, WikiCS, and PubMed datasets. The results indicate that zero-shot transfer yields limited accuracy and F1 scores, reflecting significant domain shifts. Incorporating a small number of labeled samples in the few-shot setting markedly enhances performance, demon-

Table 4: Performance of Cross-domain Node Classification

| METHODS | CITESEER | | WIKICS | | PUBMED | |
|---|---|---|---|---|---|---|
| | **ACC↑** | **F1↑** | **ACC↑** | **F1↑** | **ACC↑** | **F1↑** |
| Zero-shot | $42.79 \pm 0.31$ | $38.42 \pm 0.25$ | $15.29 \pm 0.28$ | $4.46 \pm 0.36$ | $31.49 \pm 0.30$ | $26.81 \pm 0.29$ |
| Few-shot(k=5) | $67.41 \pm 0.35$ | $60.18 \pm 0.33$ | $61.30 \pm 0.29$ | $58.05 \pm 0.27$ | $71.40 \pm 0.32$ | $71.44 \pm 0.30$ |
| Full supervised | $77.59 \pm 0.28$ | $68.24 \pm 0.23$ | $84.62 \pm 0.21$ | $82.60 \pm 0.19$ | $93.26 \pm 0.15$ | $92.78 \pm 0.12$ |

Table 5: Component-wise Analysis of Model Performance on Node Classification

| METHOD | MODULES | CORA | PUBMED | CITESEER | WIKICS |
|---|---|---|---|---|---|
| LLM | Qwen3-8B (with prompt) | $55.35 \pm 0.41$ | $86.35 \pm 0.18$ | $51.56 \pm 0.52$ | $74.41 \pm 0.29$ |
| | Llama3.1-8B (with prompt) | $52.95 \pm 0.48$ | $72.08 \pm 0.31$ | $44.46 \pm 0.63$ | $68.21 \pm 0.35$ |
| Ours | without warmup | $88.01 \pm 0.21$ | $93.76 \pm 0.11$ | $79.31 \pm 0.28$ | $86.16 \pm 0.19$ |
| | without cross attention | $89.48 \pm 0.19$ | $94.02 \pm 0.09$ | $80.41 \pm 0.25$ | $85.65 \pm 0.22$ |
| | without constractive learning | $89.11 \pm 0.23$ | $93.74 \pm 0.13$ | $81.35 \pm 0.21$ | $84.96 \pm 0.26$ |
| | without tfidf | $88.93 \pm 0.25$ | $93.03 \pm 0.15$ | $79.94 \pm 0.29$ | $85.69 \pm 0.21$ |
| | PromptGNN-sim (Llama3B) | $90.59 \pm 0.33$ | $94.12 \pm 0.09$ | $82.76 \pm 0.22$ | $85.82 \pm 0.44$ |

strating the model's adaptability to new domains. Full supervision achieves the highest scores, highlighting the importance of domain-specific training data. Overall, these findings emphasize the challenges of cross-domain graph learning and the potential of few-shot learning to mitigate domain discrepancies.

## 4.3 ABLATION STUDY

We conduct ablation experiments to systematically evaluate the contribution of each component in our model on the node classification task. By comparing variants with different embedding methods and model configurations, we isolate the impact of key design choices and demonstrate their effectiveness in improving classification accuracy.

Table 5 demonstrate our model's significant superiority over strong LLM baselines. The analysis reveals that the cross-attention mechanism and contrastive learning are critical, as their removal incurs a substantial performance degradation, thus validating their synergistic role in our architecture. We further observe that cross-attention and contrastive learning have a more pronounced impact because they directly inject the structural information contained in the dynamic prompt into the LLM representations. Cross-attention uses structural relevance to guide how the LLM processes the prompt, while contrastive learning ensures consistency between the dynamic prompt and the node's original semantics. Consequently, removing either module makes it difficult for the LLM to effectively absorb structural neighborhood information, leading to a larger performance drop. In contrast, components such as TF-IDF only adjust local text weighting and thus have a more limited effect on overall structure–text alignment. Furthermore, as shown in Table 6, TF-IDF embeddings consistently outperform ID-based features in terms of F1 score across both GCN and GAT backbones, underscoring their efficacy in capturing salient textual features for node representation.

## 4.4 PERTURBATION RESILIENCE UNDER SPARSE GRAPH SETTINGS

In this section, we evaluate the robustness of our model under sparse graph perturbations. By systematically introducing noise or removing edges, we assess how well the model maintains performance on node classification and link prediction tasks across different datasets. The results highlight the resilience of our approach compared to baseline methods in challenging, sparse scenarios.

We evaluate our model's robustness on node classification (Cora) and link prediction (Citeseer) under varying levels of structural (edge/node dropping) and semantic (text masking) perturbations (Figs. 2 and 3). Across all settings, our model consistently and significantly outperforms baselines. Specifically, it maintains high accuracy against structural perturbations that degrade GCN's

Table 6: Performance Comparison of Node Embedding Methods on Node Classification Tasks

| Method | PromptGNN-sim(GCN) | | | | PromptGNN-sim(GAT) | | | |
|---|---|---|---|---|---|---|---|---|
| | Citeseer | | Wikics | | Citeseer | | Wikics | |
| | ACC | F1 | ACC | F1 | ACC | F1 | ACC | F1 |
| TF-IDF | 79.31±0.25 | 72.15±0.31 | 85.09±0.18 | 82.46±0.21 | 82.76±0.22 | 77.92±0.29 | 85.82±0.44 | 83.85±0.19 |
| ID | 80.41±0.22 | 74.39±0.29 | 85.48±0.17 | 83.07±0.20 | 80.88±0.26 | 76.01±0.33 | 85.39±0.18 | 82.90±0.22 |

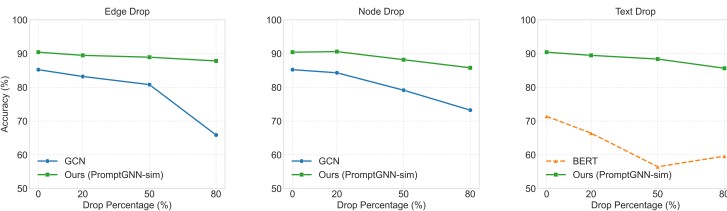

Figure 2: Cora Node Classification Robustness (ACC)

performance, while also showing greater resilience to semantic noise than BERT. These findings underscore the effectiveness of our integrated approach in learning stable and generalizable representations for noisy, real-world graphs.

### 4.5 PARAMETERS SENSITIVITY

This section analyzes how key hyperparameters affect the model's performance, providing insights into optimal settings for Cora and citeseer datasets.

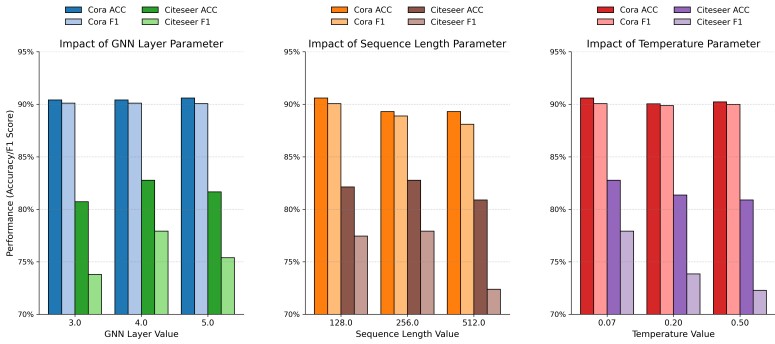

Figure 4: parameters sensitivity on cora and citeseer datasets for node classification

Figure 4 shows the effect of key hyperparameters on node classification for Cora and Citeseer. For GNN layers, Cora's accuracy and F1 remain stable around 0.90 from 3 to 5 layers, while Citeseer peaks at 4 layers with accuracy 0.8276 and F1 0.7792. Citeseer achieves best performance at sequence length 256, and Cora at 128; longer sequences slightly degrade results. A temperature of 0.07 consistently yields optimal accuracy and F1 on both datasets, suggesting sharper prediction distributions improve effectiveness.

## 5 CONCLUSION

We present PromptGNN-sim, a unified framework that enables deep bi-directional fusion between GNNs and LLMs for learning on text-attributed graphs. By combining dynamic prompt construction, cross-modal attention, and contrastive alignment, our method effectively captures both structural and semantic signals, leading to improved accuracy and generalization. Experimental results

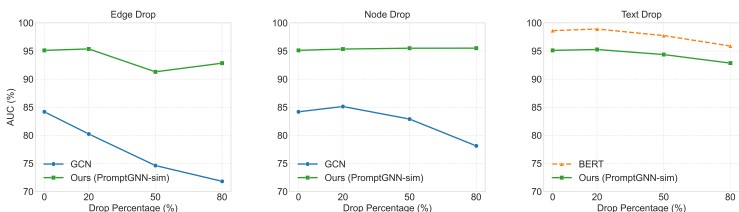

Figure 3: Citeseer Link Prediction Robustness (AUC)

across multiple benchmarks demonstrate its robustness under sparse and transfer settings. In future work, we plan to extend this framework to support inductive learning, multilingual and multimodal inputs, and scalable LLM integration.

## 6 REPRODUCIBILITY STATEMENT

To ensure the reproducibility of our results, we provide all necessary resources and detailed descriptions of the experimental setup. The source code for our models and algorithms, along with the associated training and evaluation scripts, are included in the supplementary materials, which can be downloaded directly to reproduce the experiments presented in the paper. Additionally, we have provided the processed Cora dataset results and the data generated by LLM, which are also included in the supplementary materials. Detailed descriptions of the experimental environment, hardware, and software configurations can be found in the main text and in the README file of the appendix. We encourage readers to refer to these resources to ensure that our work can be reproduced under similar conditions.

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

## A    RELATED WORK

**LLM-based Approaches for Text-Attributed Graphs.** Large Language Models (LLMs) have recently demonstrated strong capabilities in reasoning over graph-structured data by leveraging natural language prompts. For example, GraphiT Khoshraftar et al. (2025) encodes nodes and their neighborhoods into linguistically structured prompts optimized for few-shot node classification. LinkGPT He et al. (2024) reframes link prediction as a natural language reasoning task, verbalizing graph edges into prompts processed by LLMs to predict missing links, achieving competitive zero-shot and few-shot results. Related work on prompt engineering and graph verbalization Zhu et al. (2024); Yu et al. (2025) shows that LLMs can implicitly capture graph context without explicit structural modules. However, these methods lack explicit graph topology encoding, which can limit their ability to fully model relational dependencies critical for certain downstream tasks Hamilton et al. (2017); Veličković et al. (2017).

**Hybrid GNN-LLM Approaches for Text-Attributed Graphs.** To overcome the limitations of pure LLM methods, hybrid approaches integrating Graph Neural Networks (GNNs) with LLMs have gained traction. These frameworks exploit GNNs' prowess in capturing structural dependencies and LLMs' ability to model rich textual semantics. LLAGA Chen et al. (2024) integrates graph structure and language models to improve representation learning in TAGs. Similarity-based neighbor selection techniques Li et al. (2024a) enhance neighborhood quality by balancing structural and semantic signals, leading to improved model robustness. Models like LLM as GNN Zhu et al. (2025) convert graph information into a specialized vocabulary or textual sequences, allowing LLMs to implicitly learn graph context while benefiting from explicit GNN encodings. Nevertheless, current hybrid methods often rely on shallow fusion schemes Huang et al. (2024), employ static prompt designs, and struggle to maintain semantic consistency among neighbors Zhang et al. (2025). Moreover, they face difficulties in generalizing across heterogeneous graph domains and coping with noise and dynamics inherent in real-world TAGs Yan et al. (2023); Wang et al. (2025b); Lei et al. (2025).

**Robustness and Generalization in TAGs.** Robust and generalizable fusion of graph structure and textual semantics remains an open Robust and generalizable fusion of graph structure and textual semantics remains an open problem. Incomplete or noisy graph connections Zhou et al. (2020); Dai et al. (2018) and variable textual quality Wang et al. (2025a) challenge model reliability. Cross-domain shifts Li et al. (2024b); Zhang et al. (2025) further complicate learning. Prior work on contrastive learning for multimodal graphs You et al. (2020); Fang et al. (2024) and adaptive neighbor selection Li et al. (2024a) highlight promising directions, but systematic evaluation of fusion strategies under real-world perturbations is limited. Our work addresses these gaps by proposing a unified, adaptive framework enabling deep bidirectional fusion with robust cross-modal alignment and dynamic neighborhood selection.

## B    THEORETICAL ANALYSIS

In this section, we aim to demonstrate that the representations generated by LLMs from dynamic prompts can provide valuable causal features for node classification, while mitigating confounding effects. This conclusion relies on two key assumptions:

1. **Causal Fidelity**: The dynamic prompt captures the causal signal from the node's text and its filtered local neighborhood.

2. **Confounder Independence**: The dynamic prompt depends only on node text and filtered neighborhood information, and is approximately independent of potential confounders.

We formalize the theorem as follows:

**Theorem 1 (Causal Effectiveness of Dynamic Prompts under Confounder Independence).**

Given the following conditions:

1. **Causal Fidelity:** Let the dynamic prompt $P(T, L)$ be a function of node text $T$ and local neighborhood information $L$. The LLM output representation $Z_P = \text{LLM}(P(T, L))$ satisfies

$$H(Y \mid T, L) - H(Y \mid Z_P) = \epsilon, \quad \epsilon > 0 \tag{14}$$

indicating that $Z_P$ effectively preserves causal information relevant to the target $Y$.

2. **Confounder Independence:** The dynamic prompt does not depend on confounding factors $C$, i.e.,

$$I(Z_P; C) \approx 0 \tag{15}$$

Then, it follows that:

$$H(Y \mid Z_P) < H(Y \mid T) \tag{16}$$

where $Y$ denotes the node label, and $H(\cdot|\cdot)$ is conditional entropy.

**Proof.**

We aim to show that using the dynamic prompt representation $Z_P$ reduces the conditional entropy of the target $Y$, demonstrating that the prompt captures causal signals while minimizing confounding noise.

**Step 1: Represent the prompt as a function of node information.**

Since $Z_P = \mathrm{LLM}(P(T, L))$, by the Data Processing Inequality (DPI):

$$I(Y; Z_P) \leq I(Y; T, L), \tag{17}$$

i.e., any subsequent processing cannot increase the information about the target $Y$.

**Step 2: Leverage causal fidelity.**

By assumption 1, the dynamic prompt and LLM representation $Z_P$ preserve almost all causal signal:

$$H(Y \mid Z_P) = H(Y \mid T, L) - \epsilon \tag{18}$$

where $\epsilon > 0$ is a small approximation error.

**Step 3: Leverage confounder independence.**

Because the dynamic prompt does not depend on confounders $C$, and the local neighborhood is filtered based on semantic similarity and GAT attention weights, the neighborhood information is primarily relevant to the node and approximately independent of confounders:

$$I(Z_P; C) \approx 0 \tag{19}$$

This implies that the prompt representation is largely free from confounding signals.

**Step 4: Combine steps 2 and 3 to reach the conclusion.**

By properties of information theory:

$$H(Y \mid Z_P) = H(Y \mid T, L) - \epsilon < H(Y \mid T) \tag{20}$$

Hence, the LLM representation $Z_P$ generated from the dynamic prompt **preserves causal signals while avoiding confounding factors**, theoretically improving the robustness and effectiveness for node classification. While the analysis is conceptual, we note that the empirical results in Table 5 and Table 9 jointly support our theoretical assumptions: whenever components responsible for conveying the dynamic prompt's structural–semantic signal are removed or weakened, model performance drops markedly. This consistent degradation provides indirect yet strong empirical validation of our theoretical claims.

∎

## C   ALGORITHM

Algorithm 1: Model Training Procedure

---

**Algorithm 1** Model Training Procedure

---

**Require:**
1: Graph $G = (V, E)$ with node features $x_{\text{embed}}$ and texts $x_{\text{texts}}$
2: LLM-generated prompts $x_{\text{prompts}}$
3: Ground-truth labels $Y$
4: Trainable model parameters $\Theta$
5: Hyper-parameters: Learning rate $\eta$, regularization weight reg
**Ensure:**
6: Optimized model parameters $\Theta$
7: Initialize model parameters $\Theta$
8: **while** $\Theta$ has not converged **do**
9:    **for** each batch $(x_{\text{embed\_b}}, x_{\text{texts\_b}}, \dots)$ **do**
     $\triangleright$ Iterate over batches of training data
10:      $(\text{predict}, \text{loss}_{\text{reg}}) \leftarrow \text{node\_predict}(x_{\text{embed\_b}}, x_{\text{texts\_b}}, \dots, \text{reg})$
     $\triangleright$ 1. Get model predictions and regularization loss
11:      $\text{loss}_{\text{task}} \leftarrow \text{CalculateTaskLoss}(\text{predict}, \text{labels\_b})$
     $\triangleright$ 2. Calculate the primary task loss
12:      $L_{\text{total}} \leftarrow \text{loss}_{\text{task}} + \text{loss}_{\text{reg}}$
     $\triangleright$ 3. Combine losses
13:      $\Theta \leftarrow \Theta - \eta \nabla_{\Theta} L_{\text{total}}$
     $\triangleright$ 4. Update parameters via gradient descent
14:    **end for**
15: **end while**
16: **return** $\Theta$

---

## D  DATASETS

Table 7: Summary of Datasets for Node Classification and Link Prediction Tasks.

| Domain | Dataset | Nodes | Edges | Classes | Description |
|---|---|---|---|---|---|
| Citation Networks | cora | 2,708 | 5,429 | 7 | Paper titles and abstracts (category, citation) |
| | citeseer | 3,186 | 4,277 | 6 | Paper titles and abstracts (category, citation) |
| | pubmed | 19,717 | 44,338 | 3 | Medical research papers (category, citation) |
| E-commerce Networks | history | 41,551 | 358,574 | 12 | Item titles and reviews (user-item interactions) |
| | photo | 48,362 | 500,928 | 12 | Item titles and reviews (user-item interactions) |
| Knowledge Graphs | wikics | 11,701 | 215,863 | 10 | Wikipedia entries and links (knowledge graph) |

We conduct experiments on several widely used benchmark datasets shown in table 7 for graph learning, including:

- **Cora** McCallum et al. (2000): Comprising 2,708 scientific papers focused on machine learning topics, this graph captures citation relationships as edges. Papers are grouped into 7 categories. Each node includes textual content from the paper's title and abstract.

- **Citeseer** Giles et al. (1998): This dataset contains 3,186 research documents classified into 6 computer science domains. Nodes are documents, and edges indicate citation dependencies. Rich node attributes include textual summaries.

- **PubMed** Sen et al. (2008): A biomedical citation network centered on diabetes-related research, featuring 19,717 papers categorised into three medical types. The graph includes over 44,000 citation edges. Textual features derived from titles and abstracts support semantic modeling in the health domain.

- **WikiCS** Mernyei & Cangea (2020): A Wikipedia-based citation graph with 11,701 nodes representing CS-related articles and 215,863 hyperlinks as edges. The dataset is labeled into 10 computer science categories and features article text as node-level input.

- **History** Yan et al. (2023): Extracted from Amazon's book data, this graph focuses on historical literature. It includes 41,511 nodes (books) and over 300,000 co-interaction edges. Textual features consist of titles and descriptions, with classification across 12 subdomains.

- **Photo** Yan et al. (2023): From the Amazon Electronics segment, this dataset models 48,362 products and their behavioral relationships (e.g., co-purchase). Nodes are enriched with user-generated reviews, and each product falls into one of 12 categories.

## E  BASELINES

We consider the following baseline models for comparison in the experiments:

- **GCN** (Graph Convolutional Network): A spectral-based graph neural network that aggregates feature information from neighboring nodes using normalised adjacency matrices Kipf (2016).

- **MLP** (Multi-Layer Perceptron): A standard feedforward neural network applied independently to each node without considering graph structure. Rumelhart et al. (1986)

- **GraphSAGE** (Graph Sample and Aggregate): An inductive GNN that learns node embeddings by sampling and aggregating features from a node's neighborhood Hamilton et al. (2017).

- **GAT** (Graph Attention Network): Introduces self-attention mechanisms to weigh the importance of neighboring nodes when aggregating features Veličković et al. (2017).

- **NodeFormer**: A transformer-based model for graphs that captures both local and global dependencies using learned attention over node-pairs. Wu et al. (2022)

- **GLNN** (Graph Linear Neural Network): A simple yet efficient GNN that uses linear layers with graph propagation to avoid over-smoothing. Zhang et al. (2021)

- **GraphCL** (Graph Contrastive Learning): A self-supervised learning framework that enhances graph representations by maximising agreement between augmented views of graphs You et al. (2020).

- **Graphormer**: A transformer architecture designed for graph data, incorporating structural encoding like centrality and shortest path for improved performance Ying et al. (2021).

- **BERT**: A pre-trained language model based on transformers, originally developed for natural language understanding tasks Devlin et al. (2019).

- **Sentence-BERT**: A modification of BERT optimised for producing semantically meaningful sentence embeddings using Siamese and triplet networks Reimers & Gurevych (2019).

- **RoBERTa**: A robustly optimised version of BERT that improves pretraining procedures and achieves stronger performance on downstream tasks Liu et al. (2019).

- **LLAGA** Chen et al. (2024) proposes a Large Language and Graph Assistant that integrates large language models with graph structures to enhance reasoning and information extraction on textual graphs.

- **ENGINE** Zhu et al. (2024) presents techniques for efficient tuning and inference of large language models applied on textual graphs, aiming to improve scalability and performance.

- **UltraTAG-S** Zhang et al. (2025) is a unified framework that leverages LLM-enhanced text propagation and node selection techniques to address text and edge sparsity in real-world TAG learning.

- **OFA** Liu et al. (2023) proposes training a unified graph model capable of handling multiple classification tasks, moving towards a versatile and generalizable approach in graph learning.

- **PromptGFM** Zhu et al. (2025) is a recent TAG model that integrates GNN-style structural reasoning into LLMs through graph-understanding prompts and a unified graph vocabulary. It serves as a strong SOTA baseline with solid cross-graph and cross-task generalization.

- **GraphPrompte** Liu et al. (2024) aligns graph information with LLMs by using a GNN to encode graph structure and soft prompts to bridge the graph–text gap, enabling effective node classification and link prediction on TAGs.

## F  PROMPT DESIGN.

Table 8: Dynamic Prompts for Classification Across Datasets[1]

| Dataset | Prompt |
|---|---|
| cora | Classification Prediction:
Abstract: {abstract}
Title: {title}, {dynamic components}
Key References: {reference}
Graph Structure: 1.Connections: {node degree} 2.Key Neighbors: {node info}
Common Research Themes: {common keywords}
Question: Which of the following sub-categories does this paper belong to: {label}? |
| citeseer | Classification Prediction:
Abstract: {abstract}
Title: {title}, {dynamic components}
Key References: {reference}
Graph Structure: 1.Connections: {node degree} 2.Key Neighbors: {node info}
Common Research Themes: {common keywords}
Question: Which of the following sub-categories does this paper belong to: {label}? |
| pubmed | Classification Prediction:
Abstract: {abstract}
Title: {title}, {dynamic components}
Key References: {reference}
Graph Structure: 1.Connections: {node degree} 2.Key Neighbors: {node info}
Common Research Themes: {common keywords}
Question: Which of the following sub-categories does this paper belong to: {label}? |
| history | Classification Prediction:
Abstract: {abstract}
Title: {title}, {dynamic components}
Key References: {reference}
Graph Structure: 1.Connections: {node degree} 2.Key Neighbors: {node info}
Common Research Themes: {common keywords}
Question: Which of the following sub-categories does this paper belong to: {label}? |
| wikics | Input: Node and Neighbor Information: {all texts}
Common Topics: {common keywords}, {dynamic components}
Key References
Graph: Connections: {node degree}, {Key Neighbors}
Output: Question: Which of the following sub-categories of AI does this paper belong to: {label}?
Please comprehensively consider the information from the article and its neighbors, provide a comma-separated list ordered from most to least related, and only return the categories words without other words. |
| photo | Input: Node and Neighbor Information: {all texts}
Common Topics: {common keywords}, {dynamic components}
Key References
Graph: Connections: {node degree}, {Key Neighbors}
Output: Question: Which of the following sub-categories of AI does this paper belong to: {label}?
Please comprehensively consider the information from the article and its neighbors, provide a comma-separated list ordered from most to least related, and only return the categories words without other words. |

Table 8 summarizes the prompt templates used for different datasets, detailing how textual and graph-structural information—such as abstracts, titles, key references, node degrees, and common research themes—are incorporated into the input. These carefully designed prompts enable the model to leverage both node attributes and neighborhood context effectively for accurate classification across diverse domains.

Table 9: Node Classification Performance for Different Prompt Designs on the Citeseer Dataset

| Description | Prompt | Citeseer | |
| --- | --- | --- | --- |
| | | Acc | F1 |
| Abstract first | Classification Prediction: Abstract: {abstract} Title: {title} Question: Which of the following sub-categories does this paper belong to: {label}? | 81.66±0.21 | 74.12±0.28 |
| Title first | Classification Prediction: Title: {title} Abstract: {abstract} Question: Which of the following sub-categories does this paper belong to: {label}? | 80.72±0.24 | 73.82±0.31 |
| Our prompt topics | Classification Prediction: Abstract: {abstract} Title: {title}, {dynamic components} Key References: {neighbor info} Graph Structure: 1. Connections: {node degree} 2. Key Neighbors Common Research Themes: {common keywords} Question: Which of the following sub-categories does this paper belong to: {label}? | 82.76±0.19 | 77.92±0.23 |

Table 9 compares node classification performance on Citeseer using different prompt designs with Llama 3.1-8B and GPT-4o. Our comprehensive prompt, which incorporates textual content alongside graph structural features and neighbor information, achieves the highest accuracy and F1 scores, demonstrating the benefit of integrating rich contextual information in prompt construction.

## G  MORE EXPERIMENTAL DETAILS

### G.1  EXPERIMENTAL ENVIRONMENT

Our experiments were primarily conducted on a single NVIDIA GeForce RTX 4070 Ti SUPER GPU with 16GB of VRAM. For model implementation, we used PyTorch 2.5.1, CUDA 12.4, and PyTorch Geometric 2.6.1. In addition, the majority of the experiments were executed on NVIDIA A100 80GB GPUs to ensure efficient handling of larger models and datasets.

### G.2  HYPERPARAMETERS

Table 10: Hyperparameters For All Datasets

| Parameters | CORA | Citeseer | Pubmed | Wikics | History | Photo |
|---|---|---|---|---|---|---|
| warmup epochs | 20 | 20 | 20 | 20 | 20 | 20 |
| lr | $1 \times 10^{-5}$ | $1 \times 10^{-5}$ | $1 \times 10^{-5}$ | $1 \times 10^{-5}$ | $1 \times 10^{-5}$ | $1 \times 10^{-5}$ |
| weight_decay | $5 \times 10^{-8}$ | 0 | $5 \times 10^{-8}$ | $5 \times 10^{-8}$ | $5 \times 10^{-8}$ | $5 \times 10^{-8}$ |
| batch_size | 128 | 128 | 128 | 128 | 128 | 128 |
| reg | 0.1 | 0.1 | 0.1 | 0.2 | 0.1 | 0.1 |
| seqlen | 128 | 256 | 512 | 256 | 512 | 512 |
| temperature | 0.07 | 0.07 | 0.1 | 0.07 | 0.07 | 0.07 |
| neg_k | 1 | 1 | 1 | 1 | 1 | 1 |
| fixed_length | 20 | 20 | 20 | 20 | 20 | 20 |
| epochs | 30 | 50 | 30 | 30 | 30 | 30 |
| gnn.in_channels | 1433 | 384 | 500 | 300 | 384 | 384 |
| gnn.hidden_channels | 512 | 512 | 512 | 512 | 512 | 512 |
| gnn.out_channels | 384 | 384 | 384 | 384 | 384 | 384 |
| gnn.heads | 3 | 3 | 3 | 3 | 3 | 3 |
| gnn.num_layers | 5 | 4 | 4 | 4 | 4 | 4 |
| dataset.seed | 0 | 42 | 0 | 0 | 0 | 0 |
| gcn.in_channels | 1433 | 384 | 500 | 300 | 384 | 384 |
| gcn.hidden_channels | 512 | 512 | 512 | 512 | 512 | 512 |
| gcn.out_channels | 384 | 384 | 384 | 384 | 384 | 384 |

Table 10 details the hyperparameter configurations used across all datasets. Consistent settings such as learning rate, batch size, and model architecture parameters ensure fair comparisons, while dataset-specific adjustments, like sequence length and regularization strength, optimize performance for each domain. This systematic tuning supports robust and reproducible experimental evaluation.

## G.3 COMPARATIVE RESULTS

Table 11: Node classification task macro-f1

| Group | Category | Model | CORA | PUBMED | CITESEER | WIKICS | HISTORY | PHOTO |
|---|---|---|---|---|---|---|---|---|
| Baselines | GNN | GCN | 84.31±0.21 | 81.88±0.24 | 69.01±0.33 | 78.04±0.27 | 23.97±0.51 | 60.36±0.42 |
| | | MLP | 72.34±0.31 | 88.04±0.16 | 71.31±0.30 | 77.83±0.28 | 34.07±0.48 | 62.50±0.39 |
| | | GraphSAGE | 83.95±0.22 | 81.80±0.25 | 68.00±0.35 | 78.10±0.26 | 27.32±0.50 | 68.67±0.36 |
| | | GAT | 84.95±0.20 | 78.72±0.28 | 70.03±0.31 | 79.25±0.24 | 45.68±0.41 | 77.50±0.29 |
| | | NodeFormer | 77.35±0.28 | 78.74±0.29 | 69.08±0.34 | 78.23±0.27 | 22.56±0.53 | 63.48±0.40 |
| | | GLNN | 85.52±0.19 | 80.79±0.26 | 67.29±0.37 | 78.36±0.26 | 23.63±0.52 | 80.14±0.27 |
| | | GraphCL | 86.28±0.18 | 81.89±0.24 | 68.90±0.34 | 76.53±0.29 | 22.25±0.54 | 60.71±0.41 |
| | | Graphormer | 82.69±0.24 | 76.22±0.31 | 65.70±0.39 | 73.83±0.32 | 20.95±0.55 | 74.32±0.33 |
| | BERT | BERT | 69.26±0.34 | 93.26±0.11 | 67.61±0.36 | 81.07±0.21 | 49.93±0.39 | 63.95±0.38 |
| | | Sentence-BERT | 72.36±0.30 | 85.06±0.20 | 68.19±0.35 | 75.10±0.30 | 31.95±0.49 | 59.24±0.43 |
| | | ROBERTA | 72.18±0.31 | 93.64±0.10 | 68.06±0.36 | 81.52±0.20 | 54.04±0.36 | 68.35±0.37 |
| | SOTAs | ENGINE | 87.61±0.17 | 91.83±0.13 | 73.95±0.28 | 78.13±0.27 | 49.64±0.40 | 75.88±0.31 |
| | | OFA | 69.00±0.35 | 27.24±0.61 | 45.25±0.58 | 75.27±0.30 | 83.58±0.22 | 81.04±0.25 |
| Ours | PromptGNN-sim (Llama3B) | | 90.06±0.15 | 93.59±0.10 | 77.92±0.23 | 83.85±0.18 | 51.41±0.38 | 83.27±0.22 |

Table 11 shows that our GNN+LLaMA3B model outperforms baselines and state-of-the-art methods on multiple node classification benchmarks, validating the benefit of combining graph and language models.

---

[1]{label} contains candidate class names (not ground-truth labels), and the predicted label in the prompt output is generated by the LLM.

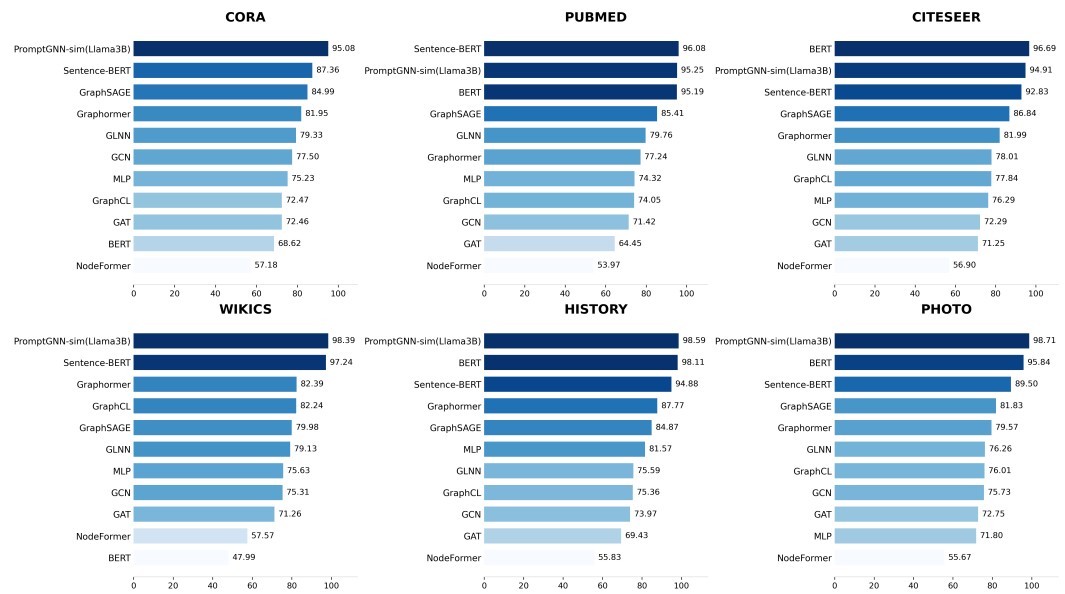

Figure 5: Link Prediction Performance: Average Precision

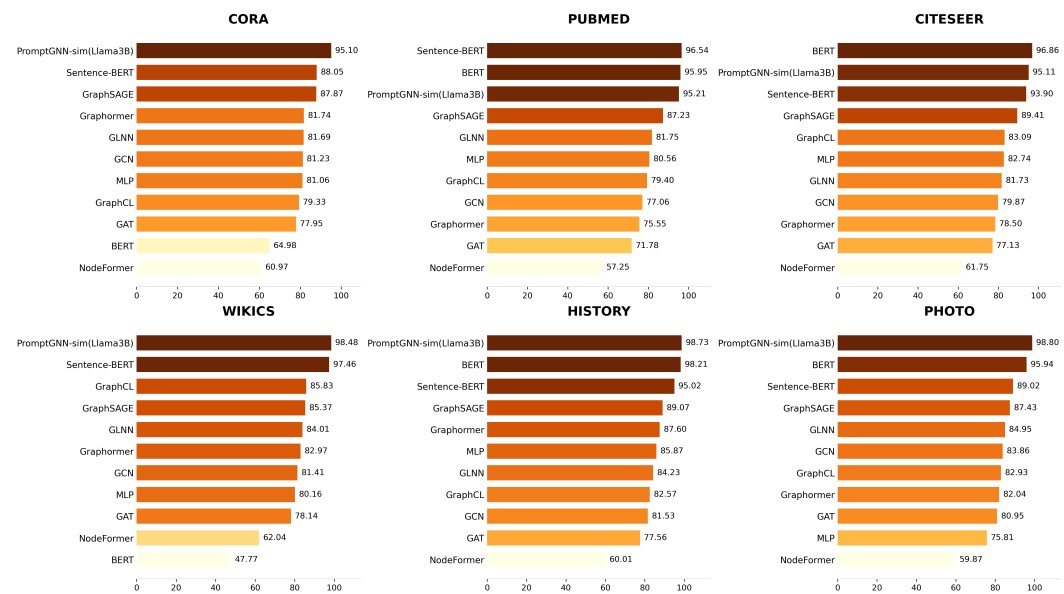

Figure 6: Link Prediction Performance: ROC-AUC

Figure 6 presents ROC-AUC results on link prediction tasks across multiple datasets. Our GNN combined with LLaMA3B consistently achieves superior performance compared to both traditional GNNs and BERT-based models, demonstrating the effectiveness of integrating graph and language models for enhanced link prediction.

Figure 5 reports average precision (AP) scores for link prediction across various datasets. Our GNN integrated with LLaMA3B consistently surpasses both conventional GNN models and BERT-based baselines, highlighting the advantage of combining graph and language models to improve link prediction performance.

## G.4 ROBUSTNESS EVALUATION

Table 12: Robustness comparison under sparse perturbations — Cora node classification (ACC).

| Method | Baseline | Drop (20%) | | | Drop (50%) | | | Drop (80%) | | |
|---|---|---|---|---|---|---|---|---|---|---|
| | | Edge | Node | Text Mask | Edge | Node | Text Mask | Edge | Node | Text Mask |
| GCN | 85.23±0.21 | 83.21±0.25 | 84.31±0.23 | - | 80.81±0.28 | 79.15±0.31 | - | 65.86±0.45 | 73.24±0.38 | - |
| BERT | 71.40±0.35 | - | - | 66.42±0.41 | - | - | 56.45±0.52 | - | - | 59.59±0.48 |
| PromptGNN-sim (Llama3B) | 90.41±0.15 | 89.48±0.18 | 90.59±0.14 | 89.48±0.19 | 88.93±0.22 | 88.19±0.24 | 88.38±0.23 | 87.82±0.26 | 85.79±0.28 | 85.66±0.29 |

Table12 presents a robustness comparison of different methods on the Cora node classification task under various sparse perturbations, including edge dropout, node dropout, and text masking at varying intensities (20%, 50%, and 80%). Our proposed framework (PromptGNN-sim(llama3b)) consistently outperforms both GCN and BERT baselines, demonstrating superior resilience to structural and textual corruptions. These results highlight the effectiveness of integrating LLMs with GNNs in maintaining performance under challenging perturbations.

Table 13: Robustness comparison under sparse perturbations — Citeseer link prediction (F1)

| Method | Baseline | Drop (20%) | | | Drop (50%) | | | Drop (80%) | | |
|---|---|---|---|---|---|---|---|---|---|---|
| | | Edge | Node | Text Mask | Edge | Node | Text Mask | Edge | Node | Text Mask |
| GCN | 70.36±0.35 | 68.20±0.39 | 70.17±0.36 | - | 65.19±0.42 | 68.10±0.39 | - | 61.80±0.48 | 67.37±0.41 | - |
| BERT | 95.58±0.11 | - | - | 95.55±0.12 | - | - | 94.09±0.15 | - | - | 89.70±0.22 |
| PromptGNN-sim (Llama3B) | 86.16±0.21 | 88.40±0.18 | 86.40±0.20 | 86.39±0.21 | 83.54±0.25 | 86.82±0.19 | 85.38±0.23 | 83.84±0.24 | 86.88±0.19 | 83.96±0.25 |

Table 13 reports the robustness evaluation of different models on the Citeseer link prediction task under sparse perturbations, including edge dropout, node dropout, and text masking at varying levels (20%, 50%, and 80%). While BERT achieves the highest baseline F1 score, our proposed method (PromptGNN-sim(llama3b)) demonstrates competitive and stable performance across all perturbation settings, often surpassing GCN. This underscores the effectiveness of our framework in maintaining link prediction accuracy under both structural and textual disturbances.

Table 14: Robustness comparison under sparse perturbations — Citeseer link prediction (ACC)

| Method | 20% Perturbation | | | 50% Perturbation | | | 80% Perturbation | | |
|---|---|---|---|---|---|---|---|---|---|
| | Drop Edge | Drop Node | Text Mask | Drop Edge | Drop Node | Text Mask | Drop Edge | Drop Node | Text Mask |
| GCN | 69.46±0.38 | 71.59±0.35 | - | 65.95±0.41 | 69.81±0.37 | - | 62.52±0.45 | 68.81±0.39 | - |
| BERT | - | - | 95.55±0.12 | - | - | 94.09±0.15 | - | - | 89.71±0.21 |
| PromptGNN-sim (Llama3B) | 88.40±0.18 | 86.51±0.20 | 86.51±0.20 | 83.55±0.24 | 86.92±0.19 | 85.26±0.22 | 83.96±0.23 | 86.98±0.19 | 84.08±0.24 |

Table 14 compares the robustness of different methods on the Citeseer link prediction task measured by accuracy (ACC) under various sparse perturbations, including edge dropout, node dropout, and text masking at 20%, 50%, and 80% intensities. While BERT attains the highest baseline accuracy, our approach (PromptGNN-sim(llama3b)) consistently maintains competitive accuracy across all perturbation settings, outperforming GCN under most conditions. These results further validate the robustness and adaptability of our integrated LLM-GNN framework in preserving predictive performance amid structural and textual disruptions.

Table 15 presents a robustness comparison of various methods on the Citeseer link prediction task measured by Average Precision (AP) across different sparse perturbations, including edge dropout, node dropout, and text masking at 20%, 50%, and 80% levels. Although BERT achieves the highest baseline AP, our method (PromptGNN-sim(llama3b)) consistently maintains high AP scores with minimal degradation under all perturbation settings, outperforming GCN in robustness. These findings further demonstrate the effectiveness of our integrated LLM-GNN framework in preserving link prediction quality despite structural and textual noise.

Table 15: Robustness comparison under sparse perturbations — Citeseer link prediction (AP). DropE, DropN, and TextM refer to perturbations on Edges, Nodes, and Text Masks, respectively.

| Method | Baseline | Drop (20%) | | | Drop (50%) | | | Drop (80%) | | |
|---|---|---|---|---|---|---|---|---|---|---|
| | | DropE | DropN | TextM | DropE | DropN | TextM | DropE | DropN | TextM |
| GCN | 85.14±0.20 | 81.56±0.25 | 85.66±0.19 | - | 76.73±0.31 | 83.09±0.23 | - | 75.56±0.34 | 78.52±0.29 | - |
| BERT | 98.49±0.05 | - | - | 98.75±0.04 | - | - | 97.63±0.08 | - | - | 95.76±0.13 |
| PromptGNN-sim (Llama3B) | 94.91±0.11 | 95.29±0.10 | 95.14±0.10 | 94.87±0.11 | 90.87±0.18 | 95.28±0.10 | 93.94±0.14 | 92.21±0.16 | 95.29±0.10 | 92.22±0.17 |

## G.5 ABLATION EXPERIMENTS

Table 16: Ablation studies on node classification task — Macro-F1 (in %).

| Method | Modules | CORA | PUBMED | CITESEER | WIKICS |
|---|---|---|---|---|---|
| LLM | qwen3-8B (with prompt) | 55.84±0.45 | 84.91±0.21 | 52.61±0.53 | 70.80±0.33 |
| | Llama3.1-8B (with prompt) | 48.84±0.51 | 56.02±0.48 | 44.59±0.58 | 60.58±0.41 |
| Ours | without warmup | 86.94±0.19 | 93.23±0.11 | 69.68±0.34 | 83.46±0.21 |
| | without cross attention | 89.16±0.16 | 93.31±0.11 | 74.50±0.28 | 83.05±0.22 |
| | without constractive learning | 88.76±0.17 | 93.09±0.12 | 73.29±0.30 | 82.37±0.24 |
| | without tfidf | 88.18±0.18 | 92.54±0.13 | 73.21±0.31 | 83.56±0.20 |
| | PromptGNN-sim(Llama3B) | 90.06±0.15 | 93.59±0.10 | 77.92±0.23 | 83.85±0.19 |

Table 16 presents ablation studies on the node classification task measured by macro-F1 scores across four datasets. The results demonstrate that each component of our model—warmup, cross-attention, contrastive learning, and TF-IDF—contributes positively to performance. Notably, the full model consistently outperforms variants with individual components removed, highlighting the effectiveness of the integrated design.

# H CASE STUDY

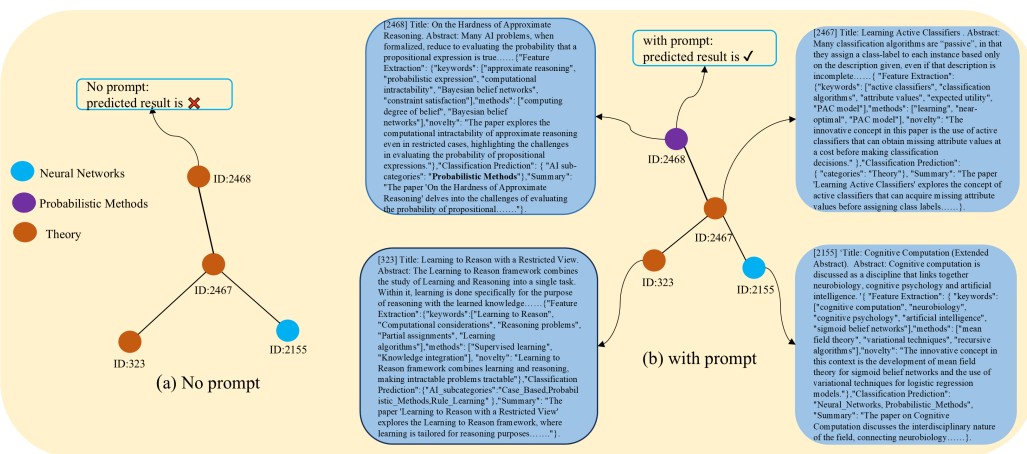

Figure 7: A case study illustration on cora dataset

This figure7 illustrates a comparison of model predictions without prompts (a) and with prompts (b). Without prompts, the model produces incorrect results due to sparse node connections and limited information propagation, which impairs its ability to capture the relationships between nodes. In contrast, when prompts are provided, the model achieves correct predictions by forming denser connections and leveraging rich contextual information around each node. This additional prompt

guidance enables the model to better understand the interactions among nodes and their neighbors, significantly improving prediction accuracy. Overall, this case study demonstrates that prompt design effectively enhances the model's focus on important features and neighborhood structures, validating its value in graph neural network tasks.

# I USE OF LARGE LANGUAGE MODELS

In this work, LLM was used primarily for text and code refinement. The LLM assisted in polishing the writing and improving the clarity of the code.

