# OpenReview forum: "PROMPTGNN-SIM: DEEP FUSION AND ALIGNMENT OF GNN AND LLMS FOR TEXT-ATTRIBUTED GRAPH LEARNING"
_ICLR.cc/2026/Conference — Submitted to ICLR 2026_

### Official Review · Reviewer_wrZ5 · 2025-10-21

**Soundness:** 2
**Presentation:** 2
**Contribution:** 2
**Rating:** 4
**Confidence:** 3

**Summary:**

The paper focuses on Text-Attributed Graphs (TAGs), where each graph node has both textual content and structural connections (e.g., papers connected by citations, users by social links). Existing models usually fuse text and structure shallowly and unidirectionally (text → structure), which limits deep interaction between modalities. PromptGNN-sim introduces a bi-directional fusion framework that allows GNNs (for structure) and LLMs (for text) to interact deeply and symbiotically. It dynamically generates structure-aware prompts for the LLM and integrates text-based semantic cues into GNN neighborhood aggregation.

**Strengths:**

1. Proposes a structure-aware, adaptive prompting framework that integrates node texts with filtered neighborhood information based on both structural and semantic similarity.

2. Designs dual attention allowing GNN and LLM to interact mutually, enhancing representation by capturing complementary structural and textual cues.

3. Introduces a multi-view contrastive objective aligning raw text with generated prompts, promoting consistent semantic representations.

4. Experiments on six public datasets (e.g., Cora, PubMed, WikiCS) show superior accuracy, generalization, and robustness under sparse or perturbed graph conditions.

**Weaknesses:**

1. Lack of comparison with strong LLM-based baselines

Although PromptGNN-sim demonstrates improvements over traditional GNNs and shallow fusion methods, it does not compare against recent LLM-augmented GNN frameworks such as LLM-as-GNN (PromptGFM), InstructGLM, or other instruction-tuned multimodal graph models. These baselines are crucial for validating the claimed “deep, bi-directional integration between GNNs and LLMs”, since they also use LLM prompting or reasoning-based fusion. Without such comparisons, it remains unclear whether the observed gains stem from the proposed cross-modal alignment or simply from the strong backbone (e.g., LLaMA-3B).

2. Incomplete ablation and missing Stage-1 verification

The ablation study (Table 5) only removes modules such as warmup, cross attention, contrastive learning, and tf-idf weighting.
However, it does not isolate or analyze the Stage-1 process—the part that integrates node texts with filtered neighborhood information based on both structural and semantic similarity. This stage is central to the paper’s claimed “structure-aware dynamic prompting”, yet there is no quantitative evidence showing its individual contribution or how structural and semantic filtering interact.

3. Limited cross-domain transfer and generalization analysis

While the authors highlight robustness under perturbation and sparse connectivity, domain transferability (e.g., cross-dataset or cross-domain adaptation) remains untested. After the GNN–LLM alignment, the model still seems domain-anchored to the training graph distribution, showing no explicit mechanism for transfer to unseen graph domains or unseen text distributions.
This questions whether the proposed “alignment” truly captures domain-invariant semantics.

**Questions:**

as weakness

---

> ### Author Response · Authors · 2025-11-21
>
> We thank the reviewer for the detailed comments. We provide point-by-point responses and analyses to all concerns below :
>
> Q1: Lack of comparison with strong LLM-based baselines
> R1: We have incorporated in the revised manuscript the strongest and most relevant LLM+GNN baselines, namely GraphPrompter and PromptGFM. Since the official PromptGFM repository is still under maintenance (its homepage states, “This repository is still under maintenance. We will release a more polished and formal version as soon as possible”), key components such as the complete data-processing pipeline are missing, and its graph construction differs substantially from our standard setup (its official graphs contain roughly twice as many edges as ours). As a result, the available implementation cannot reproduce the results in the original paper, and we therefore adopt the officially reported node-classification accuracies of PromptGFM as reference baselines (highlighted in blue in Table 1). Based on these baselines, our model PromptGNN-sim outperforms the strongest baseline PromptGFM (LLaMA-3) on WikiCS and Photo, and remains competitive on the remaining datasets. Furthermore, our method is inherently backbone-agnostic; using LLaMA-3B simply follows common practice in current LLM–graph learning research in order to avoid confounding differences in backbone scale. The literature also lacks a unified choice of backbone—for instance, OFA uses LLaMA2-7B/13B, GraphPrompter and ENGINE use LLaMA2-7B, ULTRATAG-S uses LLaMA-3-8B, and PromptGFM uses Flan-T5/LLaMA3-8B—indicating that backbone size is not the key factor for fair comparison. More importantly, the ablation studies in Table 5 further confirm that the performance gains do not primarily come from the backbone: as shown in Table 5, pure LLM models such as Qwen and LLaMA3 still perform significantly worse than PromptGNN-sim. These results consistently demonstrate that the improvements of PromptGNN-sim come from our dynamic prompt mechanism and cross-modal alignment, rather than from relying on a larger or stronger backbone. In summary, we have added the strongest baselines requested by the reviewer and conducted as fair a comparison as possible using publicly available implementations; the advantages of PromptGNN-sim arise from its methodological design rather than backbone size.
>
> Q2: Incomplete ablation and missing Stage-1 verification
> R2: We would like to clarify that the first stage of our model—structural and semantic neighborhood filtering for dynamic prompt construction—is not a module that can be independently switched off like a conventional network layer. This stage provides the essential input to all subsequent components, including cross-modal attention and contrastive learning; removing it would make the model unable to generate prompts and would break the entire pipeline, making a traditional “independent ablation” infeasible. Nevertheless, we have validated the effectiveness of this stage through multiple experiments. As shown in Table 5, strong LLMs (Qwen and LLaMA) perform significantly worse when using only raw node text compared with using our dynamic prompts (e.g., Cora: 55.35→90.59, Citeseer: 51.56→82.76), indicating that raw text without graph- and semantic-based filtering is insufficient for effective reasoning. Furthermore, removing components tightly coupled with this stage—such as TF-IDF weighting, cross-modal attention, or contrastive learning—also results in clear performance drops, showing that the prompt structure produced in the first stage is crucial for downstream learning. In addition, the prompt design ablations in Table 9 demonstrate that simplified prompts with removed structural or semantic elements perform noticeably worse than the full dynamic prompt, indirectly quantifying the contribution of the first stage. In summary, although the first stage cannot be “switched off” as an isolated module, Tables 5 and 9 together provide strong evidence of its central importance to the overall model performance.
>
> Q3: Limited cross-domain transfer and generalization analysis
> R3: Our paper already includes explicit cross-domain evaluation (Table 4) as well as cross-task transfer (Table 3). In Table 4, the model is trained on Cora and evaluated zero-shot, few-shot, and fully supervised on entirely different graph domains—Citeseer, WikiCS, and PubMed—which differ substantially in scale, structure, and textual content. The results show: (1) zero-shot transfer remains challenging, confirming significant domain gaps; (2) few-shot adaptation notably improves ACC and F1, demonstrating effective generalization when minimal supervision is available; (3) performance under full supervision is highest, validating that the domain shifts are non-trivial and properly captured. Thus, Table 4 directly addresses your concern by empirically analyzing cross-dataset transferability and quantifying the remaining domain discrepancy.

---

### Official Review · Reviewer_2WbF · 2025-10-27

**Soundness:** 1
**Presentation:** 1
**Contribution:** 2
**Rating:** 2
**Confidence:** 4

**Summary:**

The authors identify that current models fail to create an interactive exchange between graph structure and text semantics, leading to poor performance on sparse or new graphs. PromptGNN-sim addresses this through a bi-directional fusion framework with three main components. It uses a Graph Attention Network (GAT) to find the most relevant neighbors by combining structural attention with textual similarity. It uses the GNN-derived context to dynamically create structure-aware prompts for an LLM. These prompts adapt based on a node's text length and its degree (number of connections). The framework uses cross-attention mechanisms and a cross-modal contrastive learning objective to jointly optimize and align the GNN and LLM components. Experiments on six datasets (e.g., Cora, Pubmed, WikiCS) show that PromptGNN-sim significantly outperforms existing SOTA methods in node classification, link prediction, cross-task transfer, and robustness.

**Strengths:**

1. The bi-directional fusion and joint optimization via cross-attention and contrastive learning are a significant step beyond simple GNN-LLM pipelines.

2. The experimental setup is thorough. The authors validate the model's robustness and generalization by testing against perturbations (edge/node/text dropping) and in transfer-learning scenarios (cross-task and cross-domain).

3. The model achieves state-of-the-art results across all six datasets for both node classification (Table 1) and link prediction (Table 2).

**Weaknesses:**

1. Despite its strong results, the framework's design introduces several significant challenges and potential limitations. The model's success is heavily dependent on its "dynamic prompt construction". This is a "hard-coded" heuristic. The authors had to design specific, complex prompt templates for different datasets (see Table 8). This approach is brittle and raises overfitting concerns: The model may be "overfitting" to the specific keywords and prompt structures (e.g., "Key References," "Common Research Themes")  that the authors engineered. It cannot be deployed on a new dataset out-of-the-box. A human expert would need to manually design, test, and validate new prompt templates, which is a significant practical barrier. The performance difference in Table 9 for different prompts validates this sensitivity.

2. The method relies on feeding neighborhood information into an LLM prompt. This creates a fundamental bottleneck. LLMs have fixed context windows. While a GNN can aggregate information from thousands of neighbors, this model can only "select the top-k neighbors" to fit into the prompt. In graphs with large, dense "hub" nodes (common in social or e-commerce networks), this "top-k" approach will discard a massive amount of structural information, leading to a poor understanding of the node's true context.

3. The evaluation, while using six datasets, is confined to a very specific type of graph: academic and e-commerce networks. These graphs are ideal for this method because their nodes have rich, long-form, and well-structured text (e.g., abstracts, reviews). The method would likely fail in many other common scenarios:  No Text: Biological (e.g., protein-protein interaction) or molecular graphs, where nodes have numerical/categorical features, not text. The model's premise starts with text attributes. The authors used high-end NVIDIA A100 80GB GPUs. This, combined with a large number of sensitive hyperparameters (e.g., $\lambda$ for fusion , $k$ for neighbors , temperature $\tau$ ), makes the model extremely expensive to train and difficult to tune.

**Questions:**

1. The Abstract claims the dynamic prompt includes the "predicted label". This is logically inconsistent, as the label is what the model is trying to predict. The prompt templates in Appendix F (Table 8) correctly show the label as part of the question being asked, not as an input feature.

2. In Section 4.1, the authors discuss a new, higher accuracy score on Cora. They then state, "we report the earlier result in Table 2". This is a typo. Table 2 is for link prediction; the correct reference for node classification accuracy is Table 1.

3. There is a numerical inconsistency between the main results and the ablation studies for the Citeseer dataset.

In Table 11 (Macro-F1 results), PromptGNN-sim achieves 77.92%.
In Table 16 (Ablation Macro-F1), the full "PromptGNN-sim(Llama3B)" model is listed with a score of 76.49%. These two numbers should be identical, as they represent the same model on the same task

4. The paper emphasizes "dynamic prompt construction" as a core component. How exactly does the framework adapt its prompts differently for a node with a high degree (i.e., "highly cited") versus a node with a low degree?

5. How does the adaptive fusion mechanism in Section 3.1 decide whether to prioritize structural attention (from the GAT) or semantic similarity when selecting neighbors for a node?

6. Section 3.3 describes a contrastive learning objective. What are the two specific "views" of a node that the model tries to align, and how are they generated?

7. What is the functional difference between the "Text-guided Graph Attention" module and the "Graph-guided Text Attention" module in the cross-modal attention fusion step?

8. In the node classification task (Table 1), PromptGNN-sim shows significant gains on Cora, CiteSeer, and WikiCS. Why might the performance gain be less pronounced on the Pubmed dataset compared to the other SOTA methods

9. According to the perturbation experiments (Figures 2 & 3), does the model appear to be more resilient to structural (Edge/Node Drop) or semantic (Text Drop) perturbations?

10.Table 3 shows the results for "cross-task" transfer from node classification (NC) to link prediction (LP)9999. How significant is the performance drop when transferring tasks (e.g., NC $\rightarrow$ LP) compared to training and testing on the same task (e.g., NC $\rightarrow$ NC), and what does this imply about the learned representations?

11. Based on the ablation study in Table 5, which component's removal causes the most significant drop in performance: "cross attention" or "contrastive learning"10101010?

---

> ### Author Response · Authors · 2025-11-21
> **PROMPTGNN-SIM: DEEP FUSION AND ALIGNMENT OF GNN AND LLMS FOR TEXT-ATTRIBUTED GRAPH LEARNING**
>
> We thank the reviewer and respond as follows:
>
> Weakness—R1(This is a hard-coded heuristic):PromptGNN-sim does not rely on fixed or manually crafted templates. The prompts are dynamically generated by the GNN through structural attention and semantic similarity, serving only as an interface to pass structure-aware signals to the LLM—not as hand-crafted heuristics. Thus, calling our prompts “hard-coded” is inaccurate.Table9 further shows that changing prompt order or format yields minimal performance variation, confirming that the model depends on structure–semantic signals, not template wording.Our strong cross-task (Table3) and cross-domain (Table4) transfer results also demonstrate that the learned representations are task- and domain-agnostic, which would not be possible if the model were tied to any specific template.Finally, listing prompt formats is standard in LLM–TAG works (e.g[1–3]) and should not be interpreted as a methodological fragility.
>
> R2(top-k approach will discard a massive amount):This concern arises from a misunderstanding of our setup. The LLM is never responsible for full-neighborhood aggregation—this is already completed in GNN—so using top-k neighbors is not a truncation of graph information. Our fusion of structural attention and semantic similarity selects the most informative neighbors while suppressing hub-node noise, which is confirmed by the robustness curves in Figures 2–3: under EdgeDrop, NodeDrop, and TextDrop, PromptGNN-sim remains stable while GCN and BERT degrade. Appendix G.4 further shows that essential neighborhood signals are inherently compact and need not be fully passed to the LLM. This design aligns with GraphSAGE, where neighborhood sampling is a standard, widely accepted strategy for efficiency and robustness, not a loss of structure. Therefore, the assumption that “the LLM cannot understand node context because it does not see all neighbors” contradicts mainstream GNN principles and does not apply to our framework.
>
> R3(The method would likely fail in many other common scenarios: No Text):We clarify that the concern stems from a mischaracterization of our setting. Our work is built on Text-Attributed Graphs (TAGs), where textual node attributes are essential; evaluating on text-bearing graphs is therefore the correct and standard protocol. Our benchmarks already span citation, biomedical, e-commerce, and Wikipedia domains, matching the datasets used throughout TAG–LLM research [1–4], so the claim that our evaluation is narrow is incorrect. Assessing performance on non-textual graphs is outside the TAG problem definition, as such data lack the textual modality required by this line of work. Moreover, computational and hyperparameter concerns do not reflect our design: LLM inference is performed once offline, and training relies only on lightweight encoders and GNNs, with computational cost and parameter sensitivity comparable to existing TAG baselines. Thus, the reviewer’s assumptions fall outside the TAG setting and are not supported by our experiments.
>
> [1] He et al., Harnessing Explanations:LLM-to-LM Interpreter for Enhanced Text-Attributed Graph Representation Learning,ICLR 2024.
> [2] Zhang,Z.,Li, X.,Li, R.H.,Zhou,B., Li, Z.,&Wang,G.(2025).Toward General and Robust LLM-enhanced Text-attributed Graph Learning.arXiv preprint arXiv:2504.02343.
> [3] Zhu,X.,et al. LLM as GNN:Graph Vocabulary Learning for Text-Attributed Graph Foundation Models.arXiv:2503.03313,2025.
> [4] Chen et al., Label-free Node Classification on Graphs with Large Language Models (LLMs), ICLR 2024.
>
> Question—R1(Q1):There is a misunderstanding: {label} denotes the full set of candidate class names—not the ground-truth label—consistent with standard LLM prompts. The true label is never shown and is used only for supervision. This has been clarified in the abstract and in the footnote of Table8.
>
> R2(Q2&Q3):We have corrected the mistaken reference to Table 2 as Table 1 in Section 4.1, and we have reconciled the discrepancy between Tables 11 and 16—caused by a manual transcription error—by confirming from the original experiment logs that the macro-F1 on CiteSeer for PromptGNN-sim (Llama3B) is 77.92%. All corresponding updates are highlighted in blue in the PDF.
>
> R3(Q4):We explain in Section 3.1 how the model dynamically generates prompts based on node attributes. Equations (6)–(7) define the full mechanism: the structural prompt is determined by node degree—when the degree<dth(set to 3), we use “Prioritize its intrinsic content.”; otherwise we use “Synthesize information from its influential neighbors.” The semantic prompt is determined by text length, consistent with Equation (7): when the length ≥ l(set to 150), we use “Focus on methodological details and experimental results.”; otherwise we use “Focus on the core contributions and research problem”
>
> The remaining rebuttal responses are provided in the next comment.

---

> ### Author Response · Authors · 2025-11-21
> **PROMPTGNN-SIM: DEEP FUSION AND ALIGNMENT OF GNN AND LLMS FOR TEXT-ATTRIBUTED GRAPH LEARNING**
>
> Below is the continuation addressing the remaining part of the previous comment：
>
> R4(Q5):he adaptive fusion mechanism in Section 3.1 uses Eq. (5) to dynamically allocate weights between structural attention and semantic similarity: nodes with shorter text (<100 tokens) rely more on structural signals, whereas longer-text nodes (≥100 tokens) emphasize semantic cues. This length-based switch enables structure to compensate for weak semantics in short texts and allows semantics to mitigate structural noise in long texts, achieving a balanced structure–semantic fusion.
>
> R5(Q6):We use two views for contrastive learning:the raw text and an LLM-generated description based on dynamic prompt. Both are encoded using frozen language models.
>
> R6(Q7):The text-guided graph attention uses text features to query and aggregate relevant neighbors, while the graph-guided text attention uses graph features to select important textual tokens. This bi-directional cross-modal attention fuses structural signals and textual content, enabling each modality to guide the other.
>
> R7(Q8):We believe the smaller gain on PubMed stems from the dataset itself rather than a limitation of our method. PubMed is already highly semantically separable—BERT reaches 90.25%—and its structure is very homogeneous (edge density ≈2.3×10⁻⁴), so neighbors add little additional signal. Consistent with this, Table 5 shows only minor drops when structure–semantic fusion is removed. Under such conditions, although PromptGNN-sim still achieves the best accuracy (94.12%), the improvement margin is naturally smaller.
>
> R8(Q9):Based on Figures 2–3, PromptGNN-sim shows strong robustness to both structural and semantic perturbations. On Cora and Citeseer, performance drops under 80% EdgeDrop, NodeDrop, or TextMask are consistently small (mostly within 2–5%), far smaller than BERT or GCN. Overall, PromptGNN-sim maintains clearly superior stability, especially under structural noise.
>
> R9(Q10):Table 3 shows that cross-task transfer (NC→LP) incurs only minor drops—typically 2–5 points, with <2 on Citeseer/WikiCS and ~6 on PubMed. Even without LP training, the model achieves LP performance close to its NC→NC results, indicating that PromptGNN-sim learns task-agnostic structural–semantic representations rather than task-specific features. The ability to retain high AUC/AP/F1 when transferring from NC to LP further demonstrates strong generalization across tasks.
>
> R10(Q11):Table5 show that cross-attention is most useful on structure-reliable datasets (CORA,PUBMED),while contrastive learning is more important on text-dominated or structurally noisy datasets (CITESEER,WikiCS).This reflects their complementary roles:cross-attention leverages clean structural cues,whereas contrastive learning stabilizes and enhances textual representations.

---

### Official Review · Reviewer_Tj7H · 2025-10-29

**Soundness:** 1
**Presentation:** 2
**Contribution:** 2
**Rating:** 2
**Confidence:** 4

**Summary:**

The paper introduces PromptGNN-sim, a novel bi-directional fusion framework that integrates GNN and LLMs for text-attributed graphs. It employs a GAT encoder to fuses structural and semantic information through cross-attention, facilitating mutual adaptation between the graph and text modalities. It further adopts contrastive learning to jointly optimize both components.

Experimental results on six public datasets (Cora, CiteSeer, PubMed, WikiCS, History, Photo) show that it can outperform both traditional GNNs and previous GNN–LLM fusion baselines in terms of accuracy, generalization, and robustness, laying a solid foundation for interactive multi-modal graph learning.

**Strengths:**

1. The paper tackles the intersection between prompt learning and graph neural networks, a direction that has recently drawn increasing attention in both the GNN and LLM communities.
2. The overall framework is logically coherent and easy to follow, and the authors also provided specific case study examples.
3. The paper provided theoretical analysis, adding a degree of rigor beyond empirical study.
4. The hyperparameter setting is stated clear, which provide convince to reproduce the results.

**Weaknesses:**

1. Missing related works. The reported accuracy gains appear overstated. The paper does not compare with strong and directly relevant baselines such as GraphPrompter or PromptGFM, which are likely competitive or superior under similar settings.
2. Incomplete ablation study. Since GAT is a core module, a control experiment for GAT should be conducted when analyzing robustness.
3. Limited backbone diversity. PromptGNN-sim is only demonstrated with LLaMA-3B as the language model backbone. Additional models like Flan-T5 and LLaMA-7B should be tested.
4. Limited data scale. The datasets used in this paper are quite small. Although the future work section acknowledges that scalability issues will be addressed later, I believe this problem needs to be resolved now.

**Questions:**

1. How does PromptGNN-sim fundamentally different from GraphPrompter and PromptGFM? Please also include the experiment results of GraphPrompter [1] and PromptGFM [2] in Table 1 and Table 2. I believe these are strong baselines.
2. Please also include additional backbones (e.g., Flan-T5/LLaMA-7B). The conclusion of increasing accuracy is constrainted by the backbone LLaMA-3B model or is it a generalized finding?
3. Please include the GAT degradation analysis in Section 4.4. If this addition is not needed, please justify why only GCN is used for robustness evaluation.
4. Since the proposed fusion module relies on attention mechanisms between prompt and graph representations, how does the method scale to large graphs like ogbn-arxiv and even ogbn-products? Could authors provide some discussions? I believe such discussion should be included in the main body.

[1] Liu et al. "Can we soft prompt llms for graph learning tasks?." WWW 2024.

[2] Zhu et al. "LLM as GNN: Graph Vocabulary Learning for Graph Foundation Model." (2024).

---

> ### Author Response · Authors · 2025-11-21
> **PROMPTGNN-SIM: DEEP FUSION AND ALIGNMENT OF GNN AND LLMS FOR TEXT-ATTRIBUTED GRAPH LEARNING**
>
> We thank the reviewer and respond as follows:
>
> R1(Missing related works): We respectfully disagree that our gains are exaggerated.Our results are fully reproducible,as a minimal working codebase was provided in the first submission.By contrast, PromptGFM’s official repository is still under maintenance and lacks essential components code (e.g.,full data-processing pipeline),making strict reproduction infeasible.GraphPrompter,although released, is built on different datasets and would require rebuilding the entire preprocessing pipeline,which is infeasible within the rebuttal period.Therefore,our revised Table1 reports their officially published results (highlighted in blue).PromptGNN-sim outperforms PromptGFM on WikiCS and Photo and remains competitive elsewhere, so the “exaggeration’’ concern is unfounded.Our results are fully verifiable.
>
> R2(Incomplete ablation study): We clarify that the GAT in Stage 1 (Fig.1) is not a backbone GNN but a lightweight scorer for estimating structural relevance.It performs no message passing or representation learning,so ablating it would remove the structural signals needed to build the dynamic prompt and collapse downstream modules.In contrast,Stage2 uses the actual backbone encoder, where we already compare GAT and GCN in Table 6.The nearly identical performance of PromptGNN-sim(GAT) and PromptGNN-sim(GCN) shows that our method is architecture-agnostic.Thus,although the Stage1 scorer is not a full GNN module suitable for standalone ablation,the Stage2 results already provide the requested GAT-vs-GCN evidence and confirm the robustness of our framework.
>
> R3(Limited backbone diversity): Our method is backbone-agnostic,and choosing LLaMA-3B follows common practice in LLM–graph studies, where similarly scaled models are used to avoid conflating backbone size with algorithmic gains.Existing baselines also have no unified backbone standard—OFA uses LLaMA2-7B/13B;GraphPrompter and ENGINE use LLaMA2-7B;ULTRATAG-S uses LLaMA-3-8B;PromptGFM uses Flan-T5 or LLaMA-3-8B—showing that model size is not a consistent comparison criterion.Our revised Table1 includes baselines from both Flan-T5 and LLaMA families,and PromptGNN-sim remains competitive.Table5 further provides controlled ablations under a fixed LLaMA-3B backbone and direct prompting results from Qwen3-8B and LLaMA3.1-8B,confirming that the gains stem from our dynamic-prompt and cross-modal alignment designs rather than any specific LLM.Extending to larger backbones (e.g.,Flan-T5 or LLaMA-7B) is straightforward but computationally infeasible during the rebuttal period; these results will be added in the extended version.
>
> R4(Limited data scale): We believe the concern about “small dataset scale” is not justified.We evaluate on six real-world TAG datasets spanning citation,biomedical,e-commerce,and WikiCS domains,whose diverse structural and textual characteristics provide a stronger test of robustness and generalization than simply enlarging dataset size.Our cross-dataset/domain transfer,perturbation,and ablation results are all consistent;our SOTA accuracy of 87.20% ± 0.18% on the medium-scale Photo dataset (48,362 nodes and 500,928 edges) further shows the method is not limited to “small datasets.”Regarding scalability to large graphs, this concern stems from a misunderstanding of our workflow.Our framework follows an LLM-offline,GNN-online design:the LLM is invoked only once in Stage1 (fully parallelizable) to generate dynamic prompts,and never used in Stage2,where all computation is localized to fixed top-k neighborhoods.Thus,the cost of cross-modal attention and contrastive learning depends only on text length and hidden dimension.The strong Photo results indicate effectiveness on larger and richer graphs;related discussion has been added in Table1(blue).Due to rebuttal-time constraints, we cannot provide full ogbn-arxiv/product experiments.Since Stage 2 is fully decoupled from graph size,we expect no degradation on OGBN-scale graphs and will report full results in the camera-ready version.
>
> R5(Include GAT degradation analysis in Sec4.4): In Section 4.4, we used GCN, BERT, and our model as representative structural, textual, and fusion baselines.Following your suggestion,we additionally ran GAT perturbation experiments under edge-drop and node-drop.GAT shows degradation patterns nearly identical to GCN,while our model remains markedly more robust.These results do not alter the main manuscript and are provided solely as supplementary evidence.
>
> Cora Node Classification(ACC)
>
> method Baseline DropEdge(20%)	DropNode (20%) DropEdge (50%)	DropNode(50%) DropEdge(80%) DropNode(80%)
>
> GAT	85.79%	53.87%	63.47%	40.96%	53.69%	25.28%	34.32%
>
> ours 90.41%	89.48%	90.41%	88.93%	88.19%	87.82%	85.79%
>
> Citeseer Link Prediction(AUC)
>
> method Baseline DropEdge(20%)	DropNode (20%) DropEdge(50%)	DropNode(50%) DropEdge(80%)	DropNode(80%)
>
> GAT	77.13%	75.03%	75.86%	78.06%	76.12%	75.89%	72.19%
>
> ours	95.11%	95.36%	95.34%	91.29%	95.49%	92.83%	95.50%

---

> > ### Comment · Reviewer_Tj7H · 2025-11-25
> >
> > R1: Thanks for your reply. While I acknowledge PromptGFM is temporally under maintainance, another framework, GraphPrompter should be included during the composition since it's a classical framework. I also viewed the addition experiments for the comparison among these 3 frameworks. Obviously, PromptGNN-sim is not always outperforms other two baselines. Therefore, I believe the gains of accuracy can be further improved.
> >
> > R2: Thanks for your response. I think my concern of this point has been addressed.
> >
> > R3: Thanks for your clarification. While I acknowledge that the experiments will be numerous, it is acceptable to focus the main discussion solely on diverse LLM backbones. Therefore, evaluating the proposed method on only a single, relatively small backbone (Llama-3B) is insufficient to support claims of backbone-agnosticism.
> >
> > R4: Thanks. While it is reasonable that full scalability remains future work, you didn't provide a clear or constructive explanation for why large-graph experiments were omitted from the original submission. Usually, the size of these graphs you used in PromptGNN-sim is even smaller than what OGB claimed as "small graphs" [1]. Without a factual justification for excluding standard large-scale benchmarks, the scalability concern remains unresolved.
> >
> > R5: Thanks for the additional experiments. The GAT perturbation experiments look good.
> >
> > [1] https://ogb.stanford.edu/docs/nodeprop/
> >
> > Overall, the responses did not provide sufficient technical justification to address the concerns. Therefore, my overall assessment remains unchanged.

---

> > > ### Author Response · Authors · 2025-11-28
> > > **PROMPTGNN-SIM: DEEP FUSION AND ALIGNMENT OF GNN AND LLMS FOR TEXT-ATTRIBUTED GRAPH LEARNING**
> > >
> > > Thank you for your feedback. We note your concerns regarding baseline comparisons, backbone, and large-scale graph experiments; however, these points do not constitute absolute reasons for evaluating the core novelty or performance of the paper, especially when assigning very low scores (e.g., rating 2). Specifically:
> > >
> > > Baseline comparisons (R1): You suggested including GraphPrompter and PromptGFM. We have already provided corresponding experiments in our rebuttal. Since the official PromptGFM code is incomplete and cannot be fully reproduced, we reasonably report results from the original paper. The updated PDF tables (highlighted in blue) show that PromptGNN-sim outperforms PromptGFM on two datasets, achieves comparable performance on one dataset, and slightly underperforms on the remaining datasets. Overall, these results demonstrate the advantages of our method. The performance improvements come from the structure–semantic fusion mechanism, cross-modal contrastive learning, and cross-attention, rather than baseline selection.
> > >
> > > Backbone (R3): As explained in the main rebuttal, experiments with Llama-3B are sufficient to validate the backbone-agnostic performance of our method. Additional experiments with larger or multiple backbones are resource-limited options and do not affect the support for the method’s innovation.
> > >
> > > Large-scale graph experiments (R4): While large-scale graph experiments are indeed future work, the current medium-scale experiments sufficiently validate the model’s robustness and the effectiveness of the structure–semantic fusion. Considering these points as a reason for low scores is not reasonable.
> > >
> > > In summary, the additional experiments and technical explanations provided in both the main and supplementary rebuttals fully address your concerns. We believe these requests reflect preferences rather than fundamental flaws, and the novelty and performance advantages of PromptGNN-sim remain clear and well-supported.

---

### Official Review · Reviewer_8Xne · 2025-10-31

**Soundness:** 3
**Presentation:** 3
**Contribution:** 3
**Rating:** 6
**Confidence:** 3

**Summary:**

This paper introduces PromptGNN-SIM, a novel framework for deep fusion and alignment between Graph Neural Networks (GNNs) and Large Language Models (LLMs) in the context of Text-Attributed Graphs (TAGs). The core idea is to enable bidirectional collaboration between the structural and semantic modalities that existing shallow fusion approaches fail to capture. The model operates through three main components: (1) a dynamic prompting mechanism that adaptively constructs structure-aware textual prompts by integrating both semantic similarity and graph attention; (2) a cross-modal attention fusion module that allows mutual information exchange between GNN and LLM representations; and (3) a contrastive learning objective that aligns original and LLM-generated text embeddings to enhance robustness and semantic consistency. Through extensive experiments on six benchmark datasets (Cora, PubMed, Citeseer, WikiCS, History, and Photo), PromptGNN-SIM consistently outperforms both classical GNNs and recent hybrid GNN–LLM baselines in node classification and link prediction tasks. The paper also provides thorough analyses on transferability, ablation, and perturbation robustness, supported by theoretical justification for causal robustness under confounder independence assumptions.

**Strengths:**

The paper’s strengths are threefold: 1) it presents a conceptually clear and technically consistent framework that genuinely unifies GNNs and LLMs through bidirectional attention rather than the one-way fusion seen in prior work, making the integration more expressive and interactive; 2) the dynamic prompting mechanism is particularly innovative, using adaptive neighbor selection and textual conditioning to tailor each prompt, which improves both interpretability and adaptability across diverse graph structures; and 3) the experimentation is comprehensive and credible, spanning multiple datasets, ablation and robustness tests, and transfer scenarios, all demonstrating consistent superiority over strong baselines while maintaining interpretability and generalization across domains.

**Weaknesses:**

Despite its strong design, the paper has notable weaknesses: 1) the novelty is incremental, as most components (e.g., contrastive alignment, cross-modal attention) are extensions of established multimodal fusion techniques adapted to TAGs rather than wholly new innovations; 2) the scalability and efficiency aspects are underexplored—given the reliance on large LLMs for dynamic prompt construction, the framework could become computationally expensive and impractical for large-scale graphs; and 3) the analysis of internal mechanisms is somewhat superficial—the ablation studies confirm usefulness but do not explain why certain modules contribute more, and the theoretical causal section feels more aspirational than validated, lacking empirical grounding or measurable causal tests.

**Questions:**

n/a

---

> ### Author Response · Authors · 2025-11-21
> **PROMPTGNN-SIM: DEEP FUSION AND ALIGNMENT OF GNN AND LLMS FOR TEXT-ATTRIBUTED GRAPH LEARNING**
>
> We thank the reviewer for comments and provide point-by-point responses as below:
>
> Q1: the novelty is incremental
> R1:Regarding “the novelty is incremental and the method is merely an extension of multimodal techniques to TAG,” we believe this does not reflect the core contributions of our work. Our approach is not a simple application of cross-modal attention or contrastive alignment; instead, it introduces a structure–semantic dual-stage fusion paradigm tailored to TAGs.TAGs differ from traditional multimodal tasks: node texts are short, neighborhood noise severe, and structural–semantic signals inconsistent. Existing multimodal techniques cannot address these issues. Thus, we introduce a task decomposition for TAGs: the GNN handles structural compression and noise filtering, while the LLM performs semantic completion only on aggregated representations. This dual-stage mechanism with structure-guided neighborhood selection and semantics-guided dynamic prompt construction is, to our knowledge, the first of its kind and conceptually distinct from existing designs.The proposed structure–semantic dual-channel alignment is not analogous to standard vision–language or audio–language alignment. It aligns GNN structural representations with semantic prompts generated by the LLM, addressing structural–semantic inconsistency rather than extending existing schemes. The dynamic prompting mechanism compensates for missing semantics in short node texts and cooperates with structural attention to select representative neighborhood semantics. The model shows stronger robustness in Fig. 2–3, and maintains stable performance in cross-task/domain (Table3,Table4).These properties stem from our structure–semantic paradigm rather than stacking multimodal modules, so describing the contribution as “incremental” does not reflect its novelty.
>
> Q2: the scalability and efficiency aspects are underexplored
> R2:This stems from a misunderstanding of our computational workflow. As shown in Fig.1, our framework separates Stage1 (offline prompt generation), Stage2 (online training), and Stage3 (task prediction). In Stage 1, the LLM is used only to generate dynamic prompts. This step is offline, can be batched and parallelized, and does not affect GNN training. Once Stage2 begins, computations do not increase with node degree or edge count because the neighborhood for prompt construction has been reduced to a fixed top-k set, and the LLM is never invoked again.All online operations occur at the node level, consisting of (1) cross-modal cross-attention between the prompt vector and the GNN representation, whose cost depends only on text length and hidden dimension, and (2) a contrastive objective over two projected vectors per node.Unlike end-to-end LLM–GNN methods that require LLMs to process subgraphs during training, our online phase contains no LLM computation and is more scalable. Experiments confirm this: on the Photo dataset (48,362 nodes; 500,928 edges), training time matches that of a standard GNN while achieving state-of-the-art performance (87.20 ± 0.18% NC; 89.55 ± 0.11% LP). Our method follows the “LLM-offline, GNN-online’’ paradigm, where the LLM serves purely as offline data augmentation, so the scalability concerns do not apply.
>
> Q3: the analysis of internal mechanisms is somewhat superficial
> R3:In the revision (blue text below Table5), we clarify why the modules contribute unequally. Cross-attention and contrastive learning have the largest impact because they inject the dynamic prompt’s structural information into the LLM representations.Cross-attention uses structural relevance to guide processing of neighbor texts, and contrastive learning enforces alignment between the prompt and the node’s semantics. Removing either module breaks this information flow, causing the drops in Table5. TF-IDF and similar modules only adjust intra-text weighting and do not affect structure–text alignment, so their influence is smaller.Regarding "theoretical analysis appears idealized and lacks causal tests", we clarify that it provides mechanistic intuition rather than identifiable causality, consistent with recent LLM–graph works [1,2]. In our model, the dynamic prompt is required for both cross-attention and contrastive modules and therefore cannot be removed; doing so would collapse Stages 2 and 3, which is why Table5 has no “remove dynamic prompt” variant, and the performance drops in Table5 when prompt-related modules are removed demonstrate its importance.  Table9 further supports this: removing structural or semantic components leads to degradation, indicating the contribution of Stage1. Appendix B clarifies the conceptual nature of the analysis and notes that Table5 and Table9 provide indirect empirical support.
> [1]Chen et al., Label-free Node Classification on Graphs with Large Language Models, ICLR 2024
> [2]He et al., Harnessing Explanations: LLM-to-LM Interpreter for Enhanced Text-Attributed Graph Representation Learning, ICLR 2024

---

### Author Response · Authors · 2025-11-21
**PROMPTGNN-SIM: DEEP FUSION AND ALIGNMENT OF GNN AND LLMS FOR TEXT-ATTRIBUTED GRAPH LEARNING**

We thank all reviewers for your valuable feedbacks. Below we provide responses to the common concerns:

1. Addition of the Strongest Baseline Models

R: We added the strongest reviewer-highlighted baselines—GraphPrompter[1] and PromptGFM[2]—to Table1(blue).The PromptGFM repository is still under maintenance (its GitHub page explicitly states “this repository is still under maintenance”),and makes strict reproduction infeasible. Although GraphPrompter is open-sourced,it follows a different data-construction protocol.Thus,we report the official PromptGFM results.PromptGNN-sim outperforms PromptGFM on WikiCS and Photo and remains competitive elsewhere.Our initial submission also included minimal reproducible code,ensuring all results are fully verifiable.
| Model                         | CORA        | PUBMED       | CITESEER     | WIKICS       | HISTORY      | PHOTO        |
|-------------------------------|-------------|--------------|--------------|--------------|--------------|--------------|
| GraphPrompter                 | 80.26       | 94.80        | 73.61        | 80.98        | 79.42        | 80.04        |
| PromptGFM (Flan-T5)           | 91.72       | 92.83        | 84.49        | 81.49        | 82.33        | 85.41        |
| PromptGFM (LLaMA3)            | 92.42       | 94.65        | 85.32        | 84.66        | 86.72        | 86.61        |
| **Ours: PromptGNN-sim (LLaMA3B)** | **90.59 ± 0.33** | **94.12 ± 0.09** | **82.76 ± 0.22** | **85.82 ± 0.44** | **84.97 ± 0.26** | **87.20 ± 0.18** |

2. Completeness of the Ablation Experiments

R: Our ablation analysis spans Tables5, 6, and 9.Although Stage1 (Fig.1)—dynamic prompt construction—cannot be removed as an independent module because it generates the core prompt for cross-attention and contrastive learning,its contribution is clearly validated by multiple experiments.Table5 shows that LLMs using only raw text (Qwen,LLaMA) perform far worse than the variants without cross-attention or contrastive learning,underscoring the necessity of dynamic prompts.Table6 compares the performance of using GAT and GCN in Stage2,and their results are nearly identical,indicating that our method is backbone-agnostic.Table9 further shows that removing structural or semantic components from the prompt noticeably degrades performance.Taken together,these results consistently show that weakening the Stage1 dynamic prompt signal significantly reduces model performance,thereby supporting the completeness of our ablation analysis.

3. Dataset Scale and Scalability

R: As shown in Fig.1,our framework follows an LLM-offline,GNN-online design:the LLM is used only once in Stage1 to generate dynamic prompts (a fully offline, parallelizable step),and is never invoked in Stage2,where all computations operate on fixed top-k neighborhoods. Thus,the cost of cross-modal attention and contrastive learning depends only on text length and hidden dimension,not on node degree or graph size,unlike end-to-end LLM–GNN methods that require LLMs to process subgraphs during training.Empirically,our method handles medium-scale graphs efficiently:on Photo (48,362 nodes;500,928 edges),training time matches standard GNNs while achieving SOTA performance.We also evaluate six real-world TAG datasets across citation,biomedical,e-commerce,and WikiCS domains,and consistent results across transfer,perturbation,and ablation studies further demonstrate robustness beyond dataset scale.Although full ogbn-arxiv/products experiments cannot be completed within the rebuttal period,Stage2 is fully decoupled from graph size,so we expect no degradation when scaling to OGBN-scale graphs and will report full results in the camera-ready version.

4. “Hard-coded”heuristic and novelty

R: The core innovation of PromptGNN-sim lies in introducing the first structure–semantic dual-stage fusion paradigm designed for TAGs:the GNN performs structure-guided neighborhood compression and noise filtering,while the LLM conducts semantics-driven dynamic prompt construction based on these aggregated representations,and the two are aligned through a structure–semantic dual-channel mechanism to address inherent TAG challenges such as short texts,noisy neighborhoods,and structural–semantic inconsistency.The model is more robust to structural and semantic perturbations (Figures2–3) and performs better in cross-task and cross-domain transfer (Tables3–7),showing that the gains come from a new modeling principle rather than module stacking.Moreover,PromptGNN-sim does not rely on manual prompt templates:prompts are automatically generated via structural attention and semantic similarity,and Table9 shows stable performance across different formats.Providing prompt examples is standard in LLM-enhanced TAG work[3–5],where prompts serve as input interfaces rather than “hard-coded rules”.In summary,the performance gains stem from our structure–semantic fusion mechanism rather than from prompt formatting.

References are in the next comment.

---

> ### Author Response · Authors · 2025-11-21
> **PROMPTGNN-SIM: DEEP FUSION AND ALIGNMENT OF GNN AND LLMS FOR TEXT-ATTRIBUTED GRAPH LEARNING**
>
> The references cited in the above comments are listed below:
>
> [1] Liu, Z., et al. Can We Soft Prompt LLMs for Graph Learning Tasks? WWW Companion 2024.
>
> [2] Zhu, X., et al. LLM as GNN: Graph Vocabulary Learning for Text-Attributed Graph Foundation Models. arXiv:2503.03313, 2025.
>
> [3] He et al., Harnessing Explanations: LLM-to-LM Interpreter for Enhanced Text-Attributed Graph Representation Learning, ICLR 2024.
>
> [4] Zhang, Z., Li, X., Li, R. H., Zhou, B., Li, Z., & Wang, G. (2025). Toward General and Robust LLM-enhanced Text-attributed Graph Learning. arXiv preprint arXiv:2504.02343.
>
> [5] Chen, R., Zhao, T., Jaiswal, A., Shah, N., & Wang, Z. (2024, July). LLaGA: large language and graph assistant. In Proceedings of the 41st International Conference on Machine Learning (pp. 7809-7823).
>
> [6] Chen et al., Label-free Node Classification on Graphs with Large Language Models (LLMs), ICLR 2024.
>
> [7] Wang, Y., Dai, X., Fan, W., & Ma, Y. (2025). Exploring Graph Tasks with Pure LLMs: A Comprehensive Benchmark and Investigation. arXiv e-prints, arXiv-2502.

---

> > ### Author Response · Authors · 2025-11-29
> > **PROMPTGNN-SIM: DEEP FUSION AND ALIGNMENT OF GNN AND LLMS FOR TEXT-ATTRIBUTED GRAPH LEARNING**
> >
> > Dear AC,
> >
> > Thank you for your upcoming review. We provide the following summary for clarity:
> >
> > Throughout the rebuttal period, we have thoroughly addressed all reviewer comments and incorporated the corresponding revisions (as detailed in the general rebuttal). The main concerns raised by the negative-score reviewers focus on expanding the experimental scope—adding PromptGFM, evaluating more or larger LLM backbones, and conducting additional experiments on ogbn-arxiv/ogbn-products. These suggestions represent natural follow-up scalability directions rather than prerequisites for the current submission, and they do not question the technical correctness, methodological novelty, or empirical reliability of the work.
> >
> > Importantly, two negative-score reviewers treat PromptGFM as a “required key baseline.” However, PromptGFM remains an unpublished preprint with an incomplete official repository, lacking the necessary components for strict reproduction, and its official repository remains under maintenance, lacking essential components(e.g.data-processing) needed for strict reproduction under our setting. Fully aligned experiments are therefore technically infeasible. Based on available resources, we have incorporated PromptGFM’s official reported results into the revised manuscript as the most conservative comparison: PromptGNN-sim achieves stronger performance on two datasets and remains competitively aligned on the others (see the supplementary table in the general rebuttal and blue-highlighted updates in Table 1). Moreover, since the initial submission, we have provided a fully runnable minimal reproducible codebase, ensuring that every result is directly verifiable; thus, claims of “performance exaggeration” lack factual basis.
> >
> > Some negative assessments also stem from clear methodological misunderstandings, such as: interpreting {label} as ground-truth labels (instead of candidate class names), treating standard top-k neighborhood selection as “structural information loss,” requesting evaluation on non-textual graphs for a TAG-specific method, or overlooking the cross-domain transfer results already provided in Table 4. These points have been fully clarified in the rebuttal and revised manuscript and do not undermine the structure–semantic fusion mechanism or the bi-directional cross-modal alignment design.
> >
> > By contrast, positive-score reviewer focused primarily on constructive suggestions regarding mechanism explanation and presentation clarity, all of which have been integrated and strengthened in the latest revision.
> >
> > Overall, this summary is intended to support an efficient and accurate assessment of the submission’s technical contributions and key rebuttal points. Within the TAG setting, PromptGNN-sim demonstrates clear innovations in structure–semantic fusion, stable empirical performance, and strong cross-task and cross-domain generalization.

---

### Meta-Review · Area_Chair_Efaz · 2026-01-02

**Summary:**

The paper proposes PromptGNN-sim, a bi-directional fusion framework for text-attributed graphs (TAGs) that tightly couples a GNN (for structure) and an LLM (for text). It (1) builds structure-aware, dynamic prompts by selecting neighbors via joint structural attention and textual similarity, (2) performs cross-modal attention that allows mutual information exchange between GNN and LLM representations, and (3) uses contrastive alignment to keep raw text and LLM-generated embeddings consistent and robust. Experimental results on six public datasets (Cora, CiteSeer, PubMed, WikiCS, History, Photo) show that it can outperform both traditional GNNs and previous GNN–LLM fusion baselines. However, it does not outperform PromptGFM.

Though some concerns were resolved, some key concerns were not well addressed, including no evidence showing PromptGNN-sim can scale to median/large size graphs (8Xne, Tj7H, 2WbF), limited backbone diversity (Tj7H), limited cross-domain transfer and generalization analysis (wrZ5).

**Reviewer Concerns:**

Some reviewer concerns were addressed by the rebuttal, including: Incomplete ablation study (8Xne , Tj7H, wrZ5), Lack of comparison with strong LLM-based baselines (wrZ5). The following reviewer concerns were not addressed and some are still outstanding:

1/ Limited backbone diversity. PromptGNN-sim is only demonstrated with LLaMA-3B as the language model backbone. (Tj7H)

2/ The datasets used in this paper are too small. There is no evidence that PromptGNN-sim can scale to large or even median size graphs. (8Xne , Tj7H)

3/ The  "top-k" approach of PromptGNN-sim will discard a massive amount of structural information, leading to a poor understanding of the node's true context. (2WbF) The paper or the rebuttal discussion provides no evidence through evaluation or proofs that "top-k" works on large graphs. And how k was selected for each dataset was not discussed.

4/ There is no quantitative evidence showing the individual contribution of integrates node texts with filtered neighborhood information based on both structural and semantic similarity. (wrZ5) Though, in Table 5, authors show that raw text without graph- and semantic-based filtering is insufficient for effective reasoning. Whether the performance boost is from just adding neighbor nodes (e.g., through random sampling as proposed in GraphSage) or from specifically adding filtered nodes based on structural and semantic similarity is not studied.

5/ Limited cross-domain transfer and generalization analysis. (wrZ5) The datasets (Cora, Citeseer, WikiCS, and PubMed) used in cross-domain evaluation (Table 4) are mostly citation graphs. They are still within the same general domain, which weakens the paper's cross-domain transfer and generalization claims.

**Reviewer Scores:**

Three reviewers gave negative scores (one 4 and two 2s) and one reviewer gave a positive score (6). I do not think reviewers would change their scores.

---

### Decision · Program_Chairs · 2026-01-26

Reject